# A chiral fermionic valve driven by quantum geometry

Anvesh Dixit[1], Pranava K. Sivakumar[1], Kaustuv Manna[2,3], Claudia Felser[2✉] & Stuart S. P. Parkin[1✉]

Multifold topological semimetals host fermions with opposite chiralities at topological band crossings[1–3]. Chiral fermionic transport in topological systems often relies on high magnetic fields or magnetic dopants to suppress trivial transport and create an imbalance in occupancy of opposite Chern-number states[4,5]. Here we use the quantum geometry[6,7] of topological bands to filter fermions by chirality into distinct Chern-number-polarized states. This allows for the real-space separation of currents with opposite fermionic chiralities, which we have demonstrated by observing their quantum interference in the absence of any magnetic field. Devices fabricated from single-crystal PdGa in a three-arm geometry exhibit quantum-geometry-induced anomalous velocities[8,9] of chiral fermions, thereby exhibiting a nonlinear Hall effect. The resultant transverse chiral currents with opposite anomalous velocities are thereby spatially separated into the outer arms of the device. These chiral currents in opposing Chern number states also carry orbital magnetizations with opposite signs. The mesoscopic phase coherence of these chiral currents facilitated their quantum interference[10] in a Mach–Zehnder interferometer. Our findings establish a chiral fermionic valve that exhibits three key properties: spatially separates chiral fermions into Chern-number polarized states by using their quantum geometry, enables tuneable current-induced magnetization and provides a platform for controllable quantum interference of chiral quasiparticles using an electric current and magnetic field.

The chiral fermions in the topological states facilitates efficient spin and orbital angular momentum transport and play a crucial part in quantum electronics devices such as interferometers[5,10–14]. Meanwhile, chiral topological states in proximity with superconductors or magnets can lead to applications in quantum computing[15] and cryogenic memories[16]. Thus, achieving mesoscopic coherent transport of chiral fermions in these topological states is crucial to realizing low-power topological electronics[17–19]. However, access to these states in a typical topological metal is often limited because of the concurrent transport of electrons in trivial states[20]. The separation of the contributions of topological states from trivial states in their transport response is important[21,22]. Moreover, the linear electrical response of chiral fermions are innately zero because of the averaging of contributions from topological states with opposite Chern number under time-reversal symmetry (TRS)[23]. An imbalance between right-chiral and left-chiral fermions is necessary to see their transport effects in a nonmagnetic material[21]. Thus, minimizing the transport contribution of trivial states while creating an imbalance in the occupancy of right-chiral and left-chiral fermions is essential to the study of chiral fermions in topological systems. A typical strategy is through the application of large magnetic fields or by magnetic doping, which makes their practical application difficult. However, a strategy that rather filters the charge transport in the topological states from the trivial states would enable their study in the absence of any magnetic field. It would also allow for the

charge currents of opposite fermionic chirality to exist in the same device.

A non-trivial quantum geometry distinguishes the topological bands from trivial bands. The quantum geometric tensor (or Fubini–Study metric), $\mathbf{T} = \mathbf{g} - \frac{i}{2}\mathbf{F}$, consists of two parts, in which the real part $\mathbf{g}$ is the band normalized quantum metric tensor, and the imaginary part $\mathbf{F}$ is the Berry curvature tensor, related to the Berry curvature pseudovector $\mathbf{\Omega}$ as $\Omega_\alpha = \frac{1}{2}\epsilon_{\alpha\beta\gamma}\mathbf{F}_{\beta\gamma}$ (refs. 6,7,24). The use of quantum-geometry-induced phenomenon to study chiral fermionic transport in topological matter holds practical significance for two key reasons. First, it allows for the control of chiral fermionic degrees of freedom without an external magnetic field. Second, it enables access to the transport in topologically protected states, even when the transport coexists in the trivial states. Multifold topological semimetals (or chiral topological semimetals) are a unique class of materials in which their crystallographic chirality gives rise to their electronic chirality in reciprocal space[1,3,25]. The dynamics of electrons near topological band crossings are those of massless relativistic fermions with a particular chirality that is reflected in its Chern number[1,26,27]. A non-zero Chern number is closely linked with the presence of chiral topological states, for example in quantum Hall effect and Chern insulator[5,11,12]. Here we discuss PdGa, that has a P2$_1$3 space group, and whose crystallographic chirality implies the absence of any inversion centre and mirror planes. Figure 1a shows a schematic of the crystal structure of PdGa and its

[1]Max Planck Institute of Microstructure Physics, Halle (Saale), Germany. [2]Max Planck Institute for Chemical Physics of Solids, Dresden, Germany. [3]Department of Physics, Indian Institute of Technology Delhi, New Delhi, India. ✉e-mail: claudia.felser@cpfs.mpg.de; stuart.parkin@mpi-halle.mpg.de

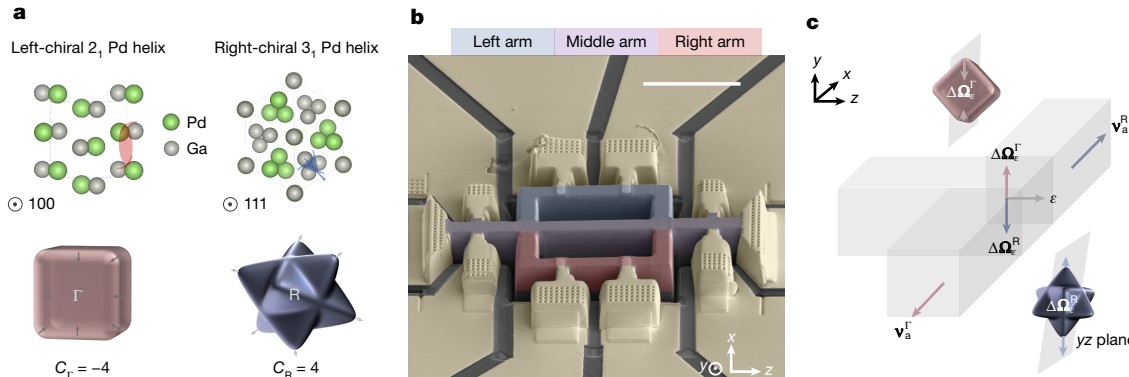

**Fig. 1 | Multifold topological semimetal PdGa. a**, Top, illustrations of the crystal structure of PdGa along the [100] and [111] directions. The schematics also show the left-chiral $2_1$ and right-chiral $3_1$ screw axes of Pd atoms along these directions. Bottom, schematic of the geometry of the Fermi pockets at the Γ and R points that have Chern numbers of $C_\Gamma = -4$ and $C_R = 4$, respectively. **b**, False-coloured SEM image of the microstructured device of PdGa made using focused-ion beam techniques, showing the three-arm geometry.

**c**, The schematic demonstrates the filtration of the charge transport in the topological states based on their non-trivial quantum geometry. The field-induced OAM $\Delta\mathbf{\Omega}_\varepsilon^\Gamma$ carried by fermions in the topological bands at Γ give them an anomalous velocity $\mathbf{v}_a^\Gamma$ towards the right arm of the device, and OAM $\Delta\mathbf{\Omega}_\varepsilon^R$ carried by fermions in the topological bands at R give them an anomalous velocity $\mathbf{v}_a^R$ towards the left arm of the device. Scale bar, 10 μm (**b**).

topological Fermi pockets $\Gamma_{FP}$ at the Γ point and $R_{FP}$ at the R point[28,29]. The signs of the Chern numbers of these pockets are related to the geometric chiralities of the screw axes along [100] and [111], respectively, in a given enantiomer. The multifold topological band crossings at the Γ and R points with opposite Chern numbers host chiral fermions with spin 3/2 (+ spin 1/2) and spin 1, respectively[1–3]. These band crossings act as the source and sink of orbital angular momentum (OAM) monopoles with large Chern numbers of ±4 (refs. 28,29). The OAM arises from the topological features captured by the Berry curvature and the spatial angular distribution of electronic states encoded in the quantum metric tensor[30,31].

In this work, we demonstrate a strategy to filter the charge transport in topological bands from trivial bands using the nonlinear Hall effect (NLH). The three-armed device geometry shown in Fig. 1b was prepared using focused-ion beam milling techniques[32,33] and designed to stimulate chiral fermionic filtration. Fermions with opposite chiralities are collected in the two outer arms of the device because of their opposite anomalous velocity, as shown in Fig. 1c. The spatial separation of fermions based on their chirality results in the preferential occupation of topological states with opposite Chern numbers in the different arms in this non-equilibrium steady state. The transport of chiral fermions in preferentially occupied topological states furthermore induces an orbital magnetization[28,30]. We observe the quantum interference of these chiral currents in the outer arms by Mach–Zehnder interferometry, which, thereby, proves their long-range phase coherence. Based on these findings, we demonstrate the working of a chiral fermionic value in which the coherent flow of chiral fermionic current can be controlled using the quantum geometry of the topological bands.

## Chiral fermionic separation by the NHL effect

Quantum transport in topological matter is influenced by the quantum geometry of the electronic bands. Nonlinear responses can thereby be used to probe the quantum geometry in transport measurements[9,34–36]. The quantum geometrical curvature of electronic bands can generate transverse currents because of the contribution from an anomalous velocity[37]. Under TRS, the anomalous velocity contribution is given by, $\mathbf{v}_a = -e\mathcal{E} \times \mathbf{\Omega}$, where $\mathcal{E}$ is the applied electric field. A non-equilibrium Berry curvature is generated in a system with non-zero **g** with an applied electric field along certain crystallographic directions[9]. This electric-field-induced Berry curvature $\mathbf{\Omega}_\varepsilon$ is given by, $\mathbf{\Omega}_\varepsilon = \nabla_k \times (\mathbf{G}\mathcal{E})$, where **G** is the Berry connection polarization tensor,

related to the **g** using the band energy derivative $\mathbf{G} = -e\frac{\partial\mathbf{g}}{\partial\varepsilon}$ (refs. 8,9,35). Here we use $\mathbf{v}_a$ as our tool for achieving charge transport filtration between the topological and trivial states. This takes place only for states with a net geometrical curvature $\Delta\mathbf{\Omega}_\varepsilon$. We denote $\Delta\mathbf{\Omega}_\varepsilon$ as the OAM dipole that arises due to the field-induced quantum geometry of the topological bands. Figure 1c shows this scheme in which fermions in topological states at $\Gamma_{FP}$ and $R_{FP}$ gain an additional transverse velocity due to the anomalous velocity induced by $\Delta\mathbf{\Omega}_\varepsilon^\Gamma$ and $\Delta\mathbf{\Omega}_\varepsilon^R$, respectively. The sign of the projection of $\Delta\mathbf{\Omega}_\varepsilon^\Gamma$ and $\Delta\mathbf{\Omega}_\varepsilon^R$ along the $y$-axis depends on the Chern numbers of $\Gamma_{FP}$ and $R_{FP}$. The $\Delta\mathbf{\Omega}_\varepsilon^\Gamma$ projection along the positive $y$-axis scatters fermions in $\Gamma_{FP}$ towards the negative $x$-axis and the $\Delta\mathbf{\Omega}_\varepsilon^R$ projection along the negative $y$-axis scatters fermions in $R_{FP}$ towards the positive $x$-axis. Therefore, at the junction of a three-arm device, as shown in Fig. 1c, fermions in the topological state at $\Gamma_{FP}$ preferentially scatter into the right arm of the device and those in the topological state at $R_{FP}$ scatter into the left arm of the device. Meanwhile, the electrons in the trivial states, which lack $\mathbf{\Omega}$, scatter preferentially into the middle arm of the device (at cryogenic temperatures when scattering events are reduced).

The transverse velocity due to $\Delta\mathbf{\Omega}_\varepsilon^{\Gamma,R}$ results in the generation of a third-order NLH response[9]. The transverse current calculated using the semiclassical Boltzmann equation is given by (Methods)

$$I_x^{\Gamma,R} \propto \int_k \partial_k(\nabla_k \times (\mathbf{G}^{\Gamma,R}\mathcal{E}_z))\mathcal{E}_z^2 \qquad (1)$$

where $\mathcal{E}_z$ is the applied electric field along $z$. The quantum geometric entity $\mathbf{G}^{\Gamma,R}$ acts as a controllable knob to filter the chiral fermions, because its value for the electronic states at $\Gamma_{FP}$ and $R_{FP}$ differ depending on the direction of $\mathcal{E}_z$ (Methods). Setting $\mathcal{E}_z = \mathcal{E}_{z,0}\sin\omega t$, the produced nonlinear transverse current can be written as a sum of first-order and third-order responses as $I_x^{\Gamma,R} = I_\omega^{\Gamma,R} + I_{3\omega}^{\Gamma,R}$ (using the identity $\sin^3\omega t = \frac{3}{4}\sin\omega t + \frac{1}{4}\sin(3\omega t + \pi)$). Their transverse voltage responses were measured under an applied longitudinal current $I_\omega$ along the $z$, aligned with the principal axis of the PdGa crystal to minimize transverse Ohmic contributions.

Several devices were fabricated out of PdGa crystal to investigate the relative contributions from $\Delta\mathbf{\Omega}_\varepsilon^\Gamma$ and $\Delta\mathbf{\Omega}_\varepsilon^R$ in the third-order NLH response. The crystallographic orientation was carefully chosen such that the transverse transport contribution from $\Delta\mathbf{\Omega}_\varepsilon^\Gamma$ and $\Delta\mathbf{\Omega}_\varepsilon^R$ are of a similar magnitude. In a typical device, the current is passed along [100] ($z$-axis), and the NLH currents are collected along [0$\bar{1}$1] ($x$-axis) (Fig. 1b). The method of synthesizing the PdGa crystals is given in ref. 38.

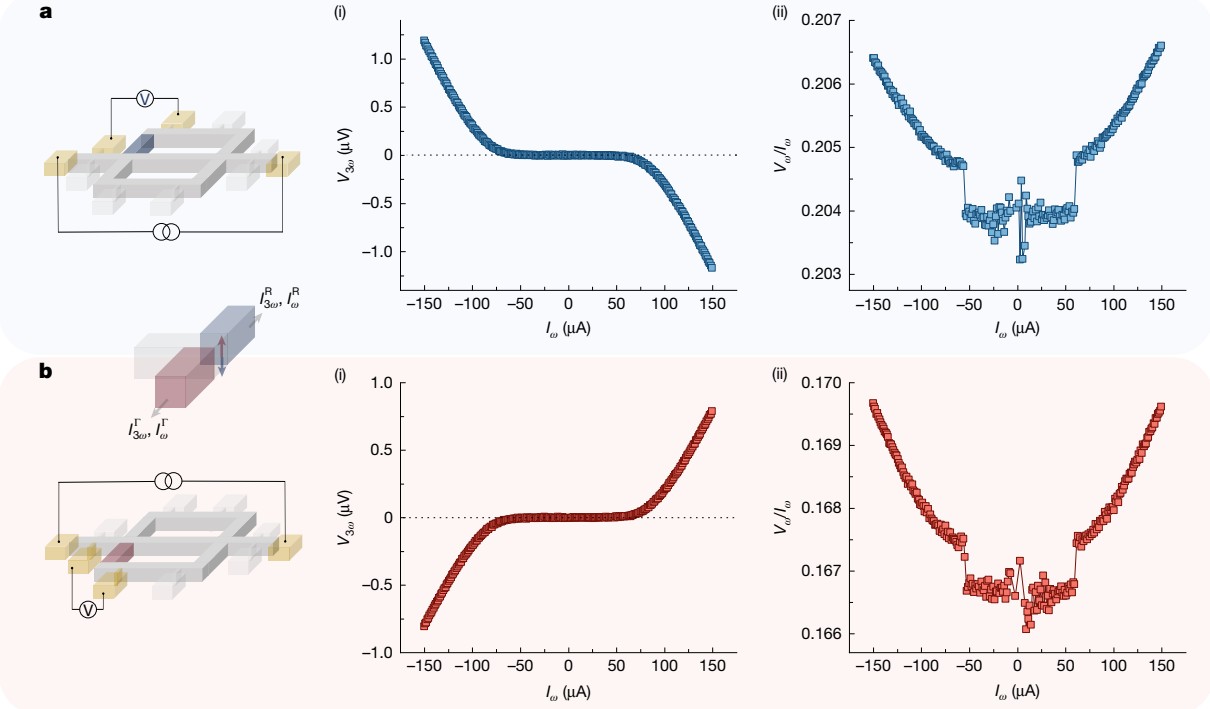

**Fig. 2 | Filtering chiral fermions from trivial charge using the NLH effect.**
**a**, Schematic of the electrical configuration for measurement of the currents $I_{3\omega}^{R}$, $I_{\omega}^{R}$ entering the left arm of the device. (i) and (ii) show the dependence of the third-order $V_{3\omega}$ and first-order $V_{\omega}/I_{\omega}$ responses on the applied current $I_{\omega}$ at 3.5 K.
**b**, Schematic showing the electrical configuration for measurement of the

currents $I_{3\omega}^{\Gamma}$, $I_{\omega}^{\Gamma}$ entering the right arm of the device. (i) and (ii) show the dependence of $V_{3\omega}$ and $V_{\omega}/I_{\omega}$ responses with respect to $I_{\omega}$ at 3.5 K. The measured $V_{3\omega}$ is the quadrature component (Y) relative to the reference signal. Although $V_{\omega}$ and $I_{\omega}$ were plotted by calculating the net magnitude of the signal (that is, $\sqrt{X^2 + Y^2}$, where X is in-phase component).

Figure 2a,b shows the dependence of the third-order NLH response $V_{3\omega}$ and the linear response $V_{\omega}/I_{\omega}$ in the left and right arms of the device, respectively, at 3.5 K, as a function of the magnitude of the applied current. Note that the schematics shown in the figures show the electrical connections made to measure these responses. As shown in Fig. 2a(i) and Fig. 2b(i), the third-order response $V_{3\omega}$ starts to appear after a threshold current of about 55 µA is reached. The opposite signs of $V_{3\omega}$ for these two cases suggests their origin is due to $\Delta\mathbf{\Omega}_{\varepsilon}^{\Gamma}$ and $\Delta\mathbf{\Omega}_{\varepsilon}^{R}$ with opposite signs of **g** (Methods). Note that the first-order response, shown in Fig. 2a(ii) and Fig. 2b(ii), remains almost constant up to 55 µA. However, a transition occurs near 55 µA, after which the first-order response starts to increase quadratically. The simultaneous appearance of third-order responses and first-order transition near 55 µA suggests that their origin is the NLH effect expected from equation (1). Notably, the first-order transition was not observed in d.c. current–voltage characteristics or d$V$–d$I$ measurements, where the resistance remains constant with the current. It further corroborates that the origin of the first-order transition is due to the NLH-induced currents. These results indicate that the nonlinear current responses $I_{\omega}^{\Gamma}$, $I_{3\omega}^{\Gamma}$ due to $\Delta\mathbf{\Omega}_{\varepsilon}^{\Gamma}$ are produced in the right arm of the device and responses $I_{\omega}^{R}$, $I_{3\omega}^{R}$ due to $\Delta\mathbf{\Omega}_{\varepsilon}^{R}$ are produced in the left arm of the device.

## Quantum metric response by a field-induced OAM

The appearance of a NLH response above a certain current threshold suggests a current-dependent **g**. This was explored from measurements of the second-order responses. Figure 3a,b shows that the $V_{2\omega}$ response starts to predominantly appear above about |55| µA, for current flowing into both the left and right arms, respectively. The $V_{2\omega}$ signal emerges simultaneously with the NLH-induced current responses shown in Fig. 2, suggesting a link between them. Notably, Fig. 3a,b also shows the sign reversal of $V_{2\omega}$ for the NLH-induced current due to $\Delta\mathbf{\Omega}_{\varepsilon}^{\Gamma}$ and $\Delta\mathbf{\Omega}_{\varepsilon}^{R}$. Recent studies have linked the occurrence of $V_{2\omega}$ due to a

quantum metric in a system that maintains $\mathcal{PT}$ symmetry while simultaneously breaking $\mathcal{P}$ and $\mathcal{T}$ (refs. 35,39). Similar symmetry arguments can be made be for $\Delta\mathbf{\Omega}_{\varepsilon}^{\Gamma, R}$, which acts like an orbital analogue of spin in a topological antiferromagnet[40]. Thus, the observed $V_{2\omega}$ responses for $\Delta\mathbf{\Omega}_{\varepsilon}^{\Gamma}$ and $\Delta\mathbf{\Omega}_{\varepsilon}^{R}$ show the presence of a non-zero **g** in our system after the current threshold. Meanwhile, the sign reversal suggests that the **g** of $\Gamma_{\mathrm{FP}}$ and $R_{\mathrm{FP}}$ have opposite signs. Therefore, as per equation (1), the non-zero **g** generates NLH-induced transverse currents in the right and left arms of the device. These findings further corroborate the role of $\Delta\mathbf{\Omega}_{\varepsilon}^{\Gamma}$ and $\Delta\mathbf{\Omega}_{\varepsilon}^{R}$ in the charge separation of fermions in topological states. Furthermore, we have confirmed that the observed NLH responses are not attributable to trivial states, despite the potential coexistence of trivial-state mechanisms with quantum geometric effects (Methods).

## Chiral fermionic current carries orbital magnetization

The chiral fermions scattered transversally due to $\Delta\mathbf{\Omega}_{\varepsilon}^{\Gamma}$ into the right arm and $\Delta\mathbf{\Omega}_{\varepsilon}^{R}$ into the left arm of the device occupy topological states at $\Gamma_{\mathrm{FP}}$ and $R_{\mathrm{FP}}$, respectively, which, thereby, preserves their chirality[41]. Thus, the transverse current generated by $\Delta\mathbf{\Omega}_{\varepsilon}^{\Gamma}$ and $\Delta\mathbf{\Omega}_{\varepsilon}^{R}$ is preferentially carried by topological states at $\Gamma_{\mathrm{FP}}$ and $R_{\mathrm{FP}}$, respectively. The resulted preferential scattering would create an imbalance in the occupation of chiral fermions in the topological states in the left and right arms, as shown schematically in Fig. 4a. This charge transport by chiral fermions in Chern-number-polarized-topological states is termed as chiral current. The occupational imbalance of these topological states allows for chiral currents to carry a finite orbital magnetization ($\mathbf{m}^{\Gamma, R}$) derived from $\Delta\mathbf{\Omega}_{\varepsilon}^{\Gamma, R}$ (ref. 21). The $\mathbf{m}^{\Gamma, R}$ carried by the chiral current follows the symmetry of the preferentially occupied Fermi pocket. The chiral currents for different applied magnetic field ($B$) orientations were studied to reveal their underlying symmetry. The modulation in the density of states by the term $\mathbf{B} \cdot \mathbf{\Omega}_{\varepsilon}^{\Gamma, R}$ directly influences the chiral currents[42]. Hence, the field orientation $\theta$-dependent modulation of the chiral current

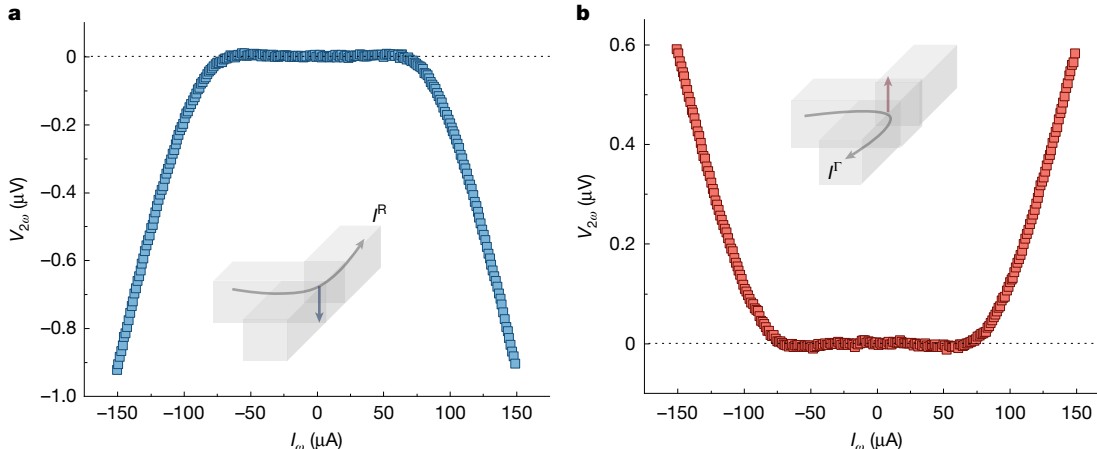

**Fig. 3 | Quantum metric due to electric-field-induced OAM. a,b,** Dependence of second-order response $V_{2\omega}$ on the applied current $I_\omega$ for current going into the left (**a**) and right (**b**) arms. The electrical connections for **a** and **b** are the same as those used in Fig. 2a,b. The measured $V_{2\omega}$ is the in-phase component of the lock-in signal.

reveals the underlying symmetry of $\Delta\mathbf{\Omega}_\varepsilon^{\Gamma,R}$. This was measured in the left and right arms using electrical contacts along these arms, as shown schematically in Fig. 4b,c. The magnetic field was rotated in the $xy$ plane, where $\theta = 0°$ corresponds to the field oriented along $y$.

The $\theta$-dependent third-order and first-order responses for the left arm are shown in Fig. 4b and compared with the same responses for the right arm in Fig. 4c. Highly distinctive responses are found for the two arms reflecting the distinctive symmetries of the respective Fermi pockets. $R_{FP}$ has a three-fold symmetry, whereas $\Gamma_{FP}$ has a four-fold symmetry (Fig. 4a). For the left arm, the inherent three-fold symmetry of $R_{FP}$ is broken by $\mathbf{G}^R\mathcal{E}_z$, which gives rise to the observed modulations around 60° and 240°. These results indicate that the topological states at $R_{FP}$ preferentially carry the chiral current in the left arm. Whereas, for the right arm, Fig. 4c(i),(ii) rather show modulations of $V_\omega$ and $V_{3\omega}$ around 90° and 270°, respectively, that is, a two-fold symmetry. These $\theta$-dependent modulations follow the symmetry of $\Delta\mathbf{\Omega}_\varepsilon^\Gamma$ in which the

four-fold symmetry of $\Gamma_{FP}$ is broken by $\mathbf{G}^\Gamma\mathcal{E}_z$. Thus, the topological bands at $\Gamma_{FP}$ preferentially carry the chiral current in the right arm. Thus, we conclude that the chiral currents in the left and right arms result from their preferential occupancy in topological states with opposite Chern numbers. Notably, the signs of the relative change in magnitudes of both $V_\omega$ and $V_{3\omega}$ are of opposite sign in the left and right arms, respectively, as can be seen in Fig. 4b,c. This shows that the orbital magnetizations $\mathbf{m}^\Gamma$ and $\mathbf{m}^R$ have opposite polarities. These results show the two key characteristics of the chiral currents in the arms of the device: they preferentially occupy Chern-number-polarized-topological states with opposite Chern numbers, and they carry orbital magnetizations with opposite polarities.

## Quantum interference of chiral current

The observed nonlinear response in our mesoscopic devices is an indicator of the long-range phase coherence of the chiral currents.

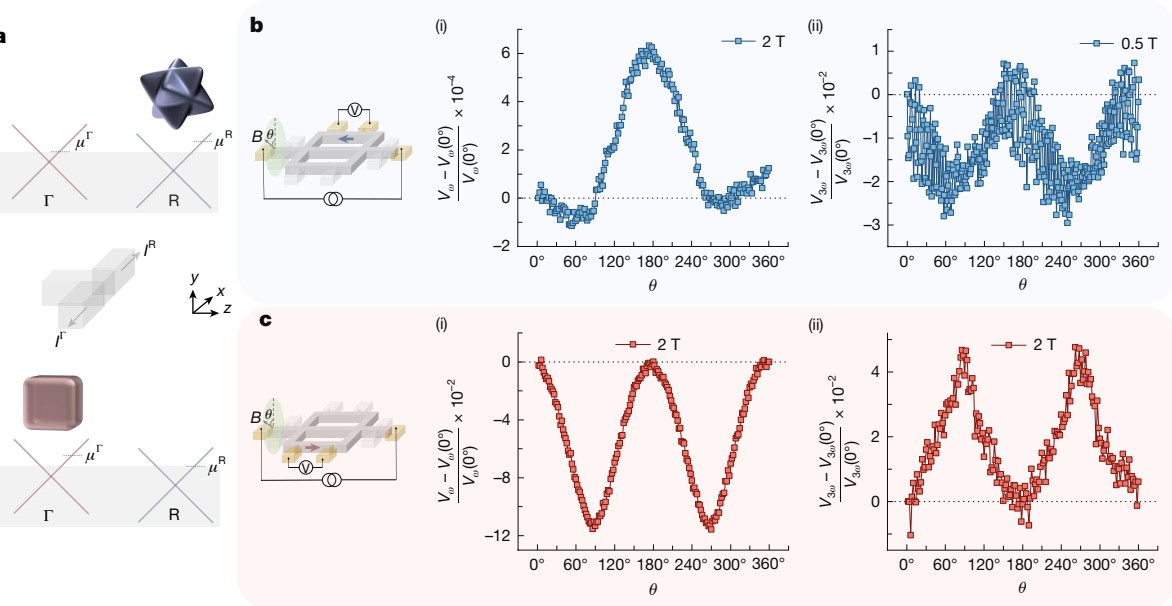

**Fig. 4 | Chiral fermionic current carries orbital magnetization. a,** Schematic showing the origin of the chiral fermionic current due to the preferential occupancy of topological bands at Γ and R in the right and left arms, respectively. **b,c,** Magnetic field orientation $\theta$ dependence of variation of first-order and third-order electrical responses normalized with respect to the responses at $\theta = 0°$, in the left and right arms, respectively. The magnetic field is rotated in the $xy$ plane, where $\theta = 0°$ corresponds to the $y$-axis. The data for **b**(i) and **c**(i) and (ii) were measured at 2 K with an applied current of 400 µA in a magnetic field of 2 T. Although the data for **b**(ii) were obtained at 50 mK at an applied current of 70 µA and magnetic field of 0.5 T.

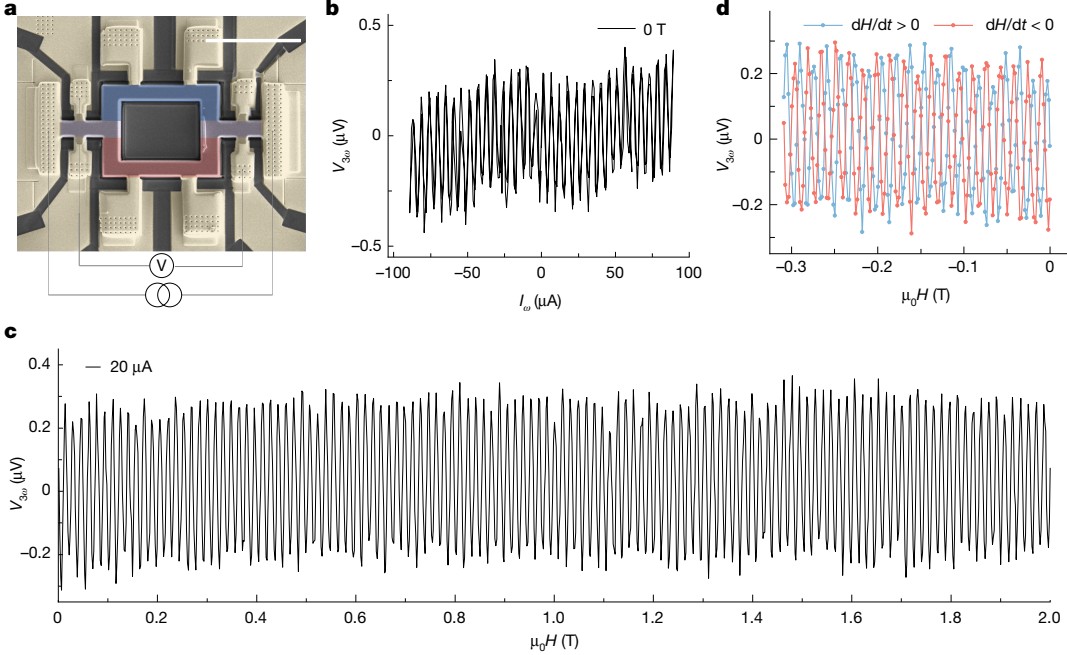

**Fig. 5 | Quantum interference of chiral current. a**, False-coloured SEM image of the MZI made from the PdGa crystal. The applied electric field is along [0$\bar{1}$1], and NLH-induced current were collected along [100]. **b**, Oscillations in $V_{3\omega}$ with applied current at 30 mK in zero external magnetic field. The current was swept in both directions towards the positive and negative polarities. **c**, Oscillations in $V_{3\omega}$ with applied magnetic field $\mu_0 H$ at 30 mK with a current of 20 μA. **d**, Dependence of the oscillations in $V_{3\omega}$ on the sweep direction of $\mu_0 H$ at 30 mK with a current of 20 μA. The magnitude of the field-sweep rate in **c** and **d** was 0.83 mT s$^{-1}$. The magnetic field was applied out of plane along the $y$-axis. Here $V_{3\omega}$ is the in-phase component of the lock-in signal. Scale bar, 10 μm (**a**).

To study this further, we prepared a Mach–Zehnder interferometer (MZI) device, as shown in Fig. 5a, that has a similar geometry to the device studied earlier but without a middle arm. The chiral currents induced by the NLH effect in the left and right arms remain the same in the MZI. However, the first-order Ohmic current will coexist with the first-order chiral current in both arms, whereas the third-order chiral currents show similar characteristics as before. The dependence of $V_{3\omega}$ measured across the MZI on the applied current at 30 mK is shown in Fig. 5b. Clear oscillations in $V_{3\omega}$ are found over the entire range of current applied. Moreover, the oscillations overlap for increasing and decreasing current.

The chiral currents $I_{3\omega}^{\Gamma}$ and $I_{3\omega}^{R}$ acquire a phase due to the current-induced orbital magnetizations $\mathbf{m}^{\Gamma}$ and $\mathbf{m}^{R}$ in the right and left arms, respectively. The phase acquired by the chiral currents have different signs because $\mathbf{m}^{\Gamma}$ and $\mathbf{m}^{R}$ have opposite polarities. Meanwhile, the absolute phase difference acquired by the chiral current varies with the applied current, because $\mathbf{m}^{\Gamma}$ and $\mathbf{m}^{R}$ are dependent on current. Thus, the phase difference carried by the chiral current results in $V_{3\omega}$ oscillating with the applied current with a well-defined period (Methods). The current-oscillation period $\Delta I \approx 6$ μA corresponds to a change in the flux linkage by a magnetic flux quantum $\phi_0$. We can calculate an effective current-to-flux conversion coefficient (or effective inductance $L_{\text{eff}}$) of the interferometer using the relations $L_{\text{eff}} = \varphi_0/\Delta I \approx 0.68$ nH. The quantum interference of the chiral currents passing through the two macroscopic arms without any applied magnetic field is possible only if, first, chiral currents of opposite chiralities exist in the left and right arms, and, second, the chiral currents have a long-range phase coherence of more than 15 μm.

The phase coherence of these chiral fermionic currents was further explored in the presence of an external magnetic field oriented along $y$. As shown in Fig. 5c, oscillations of $V_{3\omega}$ were observed, which were periodic over magnetic field. We also observed that the phase of the oscillations was shifted by π, when the magnetic field sweep direction was reversed, as shown in Fig. 5d. It shows the inductive nature of the voltage response of the chiral fermionic current (Methods). Assuming

a single quasi-free electron interference similar to the Aharonov–Bohm effect, the $B$-oscillation period of 15.82 mT gives an interference path area $A_I = \phi_0/\mu_0 \Delta H \approx 0.26$ μm$^2$. $A_I$ is about 400 times smaller than the actual device geometry. The assumption of single quasi-free electron behaviour is invalid in our case because the chiral currents consist of topological quasiparticles with spin 1/2, spin 1 and spin 3/2 having orbital magnetic momentum in different arms[1,43]. The effective $g$-factor of topological quasiparticles in systems with strong spin–orbit coupling can be significantly higher than that of a free electron[44]. Thus, the total phase accumulated due to enhanced spin-phase contribution[33] and conventional Aharonov–Bohm phase may account for the observed larger oscillation period than conventional free-electron systems. Further theoretical studies are also needed to understand the decoherence mechanism and quantify the coherence length. The influence of interacting fermions on quantum interference makes our system an interesting platform to probe topological quasiparticle excitations.

## Conclusions

Innovative ways of controlling and manipulating electronic degrees of freedom initiates advances in electronics. Semiconductors enable switchable valves for charge flow in a transistor[45], whereas magnets enable spin valves for control of electron spin flow[46]. Our work presents a new device concept of a chiral fermionic valve, which filters chiral fermions in topological states from charge transport in trivial states using their quantum geometry. Furthermore, we show the mesoscopic phase coherence of the chiral fermions through the quantum interference of their chiral currents. The geometry of the three-arm device gives rise to the filtration of chiral fermions through the NLH response driven by an electric-field-induced OAM dipole $\Delta \boldsymbol{\Omega}_{\varepsilon}^{\Gamma, R}$. The $\Delta \boldsymbol{\Omega}_{\varepsilon}^{\Gamma}$ and $\Delta \boldsymbol{\Omega}_{\varepsilon}^{R}$ induced currents exist with opposite chiralities in the right and left arms, respectively. The preferential occupancy of chiral currents in topological states was confirmed by studying their modulation on varying the external magnetic field orientation. Our results indicate that the chiral currents carry orbital magnetizations with opposite

polarities in the respective arms in Chern-number-polarized topological states. We show that the generation of orbital magnetization at the mesoscale is linked with the long-range coherence of the chiral currents. The quantum interference of chiral currents in a Mach–Zehnder interferometer was observed, which confirms their mesoscopic phase coherence. Based on these findings, we propose a chiral fermion valve in which fermions in topological bands are separated based on their chirality using their non-trivial quantum geometry. Similar to the gate voltage in a transistor, the control knob for the chiral fermionic value is the field-induced OAM (Methods). Chiral fermionic valve establishes three remarkable abilities: (1) it spatially separates chiral fermions of opposite chiralities into Chern-number-polarized topological states; (2) it enables control over both the polarity and magnitude of current-induced magnetization; and (3) it provides a versatile platform for studying the quantum interference of chiral quasiparticles, which can be tuned using electric current and magnetic field. Our findings apply to the extended family of topological materials with multifold topological crossings, enabling access to chiral topological states without magnetic fields, doping or electrostatic gating. The chiral fermionic valve opens up new device applications for harnessing the chiral degree of freedom in future quantum electronic devices.

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

# Methods

## Device preparation and electrical measurements

The procedure for device fabrication from PdGa crystal using Ga-ion-based focused-ion beam technique is as follows[32]. A lamella was milled out of the bulk crystal. Then the lamella was transferred in situ to the TEM grid for polishing till the desired thickness is achieved. The thickness of the lamella used to make devices in our study ranges from 1 μm to 4 μm. Then the lamella was microstructured into desired three-arm geometry on the TEM grid. Then the microstructured lamella was transferred in situ onto the SiO$_2$ substrate with prepatterned Au contact pads. The electrical contacts between the lamella and the contact pads were made by sputtering Ti–Au bilayers. Then the etching step was carried out to remove shorting between the electrical contacts. During the device preparation procedure, special care was taken to reduce the surface damage. For example, the transport channel in the device was never scanned with the Ga ion beam after the fine polishing step. However, surface damage of few tens of nanometres may still exists even after several fine polishing steps. Therefore, a dry etching step using low-energy Ar ion was performed to further reduce the thickness of this amorphous layer before the deposition of Ti–Au to make electrical contacts. The effect of surface damage on transport measurements was also studied by purposefully damaging the sample surface with the ion beam. The comparison of results before and after ion-irradiation were compared to study the influence of the amorphous layer on transport. As also observed in previous studies, the increase of the amorphous layer thickness did not negatively influence the transport response through topological states[33].

The electrical measurements were primarily performed in a Bluefors LD−400 dilution refrigerator. Meanwhile, experimental data in Fig. 4 with the applied magnetic field of 2 T were measured in a PPMS Dyna-Cool cryostat. The electrical measurements were carried out using Zurich Instruments lock-in amplifiers (MFLI) at 7.919 Hz and 13.333 Hz reference frequencies. The oscillator voltage amplitude was varied to sweep the applied current through the device (with a buffer resistance in series). The higher harmonic voltage responses from devices were simultaneously measured with the multi-demodulator option provided in an MFLI. Each demodulator produces two output signals: one corresponding to the in-phase component ($X$) and the other to the quadrature component ($Y$) relative to the reference signal. Special considerations were taken to increase the signal-to-noise ratio, such as using high-frequency electronic filters and avoiding ground loops. The magnetic field orientation was swept using a three-axis superconducting magnet from American Magnetics in a Bluefors refrigerator (Fig. 4b(ii)). An out-of-plane rotator puck was used in the PPMS case, in which the rotation axis is along the applied current under the magnetic field of 2 T (Fig. 4b(i),c). Two different configurations of electrode contacts with the electrical transport channel in a device were studied. It was observed that the measured third-order voltage responses were more prominent when the Ti–Au connected the top surface of the channel with the electrical contacts. The enhanced third-order response made it possible to measure the $B_\theta$ dependence of the third-order response, as shown in Fig. 4b(ii),c(ii).

We have measured 23 devices fabricated in different geometries and crystal orientations. The five focal crystal orientations are presented in Extended Data Table 1. We can define four working states of the valve, depending on the relative magnitude of the chiral currents in the two arms. We have quantified the position of the valve using the term $\phi = \tan^{-1}\frac{V_{3\omega}^{R}}{V_{3\omega}^{L}}$, as shown in Extended Data Fig. 1. $\phi = 45°$ represents 'valve on' state, when there are equal magnitudes of the NLH currents from the Γ and R Fermi pockets in the right and left arms, respectively. $\phi = 0°$ represents '$I^\Gamma$ on' state and $\phi = 90°$ represents '$I^R$ on' state, where NLH current in the right arm is predominantly generated due the Γ Fermi pocket and that in the left arm is predominantly generated due the R Fermi pocket. Finally, the 'valve off' state, in which chiral current

generation is suppressed in both arms. We made four devices in a three-arm geometry near to the 'valve on' position. All four devices show the same experimental features discussed in the paper, namely, the appearance of distinct NLH responses of similar magnitude in different arms after the current threshold and their distinct symmetries of first-order responses with magnetic field orientations, as discussed in Fig. 3b(i),c(i). The absolute value of the nonlinear response varied with the dimensions of the device, as given in the Extended Data Table 2. The modulations of the third-order responses discussed with Fig. 3b(ii),c(ii) were observed in two of these devices. A good signal-to-noise ratio is needed to measure the nonlinear responses in the presence of the magnetic field, which we posit is a limiting factor in our PPMS system. We made three Mach–Zehnder interferometer (MZI) devices with a 'valve on' crystallographic position. All three showed oscillations in third-order response with applied current and magnetic field, as discussed in Fig. 5. The interference visibility $\vartheta$ for the MZI with $\phi = 49.4°$ and 38.6° were 0.86 ± 0.06 and 0.69 ± 0.1, respectively. $\vartheta$ was calculated using the relation $\frac{V_{3\omega}^{amp} - V_{3\omega}^{avg}}{V_{3\omega}^{amp} + V_{3\omega}^{avg}}$, where $V_{3\omega}^{amp}$ is the distance between the peak and crest of the oscillation, which is calculated using $|V_{3\omega}^{max} - V_{3\omega}^{min}|$ and $V_{3\omega}^{avg}$ is the mean offset of the oscillation given by $|\frac{V_{3\omega}^{max} + V_{3\omega}^{min}}{2}|$. $V_{3\omega}^{max}$ and $V_{3\omega}^{min}$ are the minimum and maximum values of the $V_{3\omega}$ signal. We also found that $\vartheta$ was sensitive to the electronic properties of the conduction channel sidewall. The value of $\vartheta$ increased when the sidewall opposite to the voltage probe is not electrostatically screened by a presence of another electrode.

## Passive and active control of the chiral fermionic valve

The chiral currents generated in the two arms of the device depend on the magnitudes of $\Delta\mathbf{\Omega}_\varepsilon^\Gamma$ and $\Delta\mathbf{\Omega}_\varepsilon^R$, as given in equation (1). We can control the relative projection of $\Delta\mathbf{\Omega}_\varepsilon^\Gamma$ and $\Delta\mathbf{\Omega}_\varepsilon^R$ along $y$, using the distinct symmetry of the Fermi pockets at the Γ and R points. Although the symmetry of a crystal is defined by its space group, the symmetry of a Fermi pocket is determined by the local symmetry of the band structure at a given $\mathbf{k}$-point[47]. Therefore, Fermi pockets at the Γ and R points can locally exhibit distinct mirror-like symmetry $\mathcal{M}^*$, which maps $(k_x, k_y, k_z) \longrightarrow (k_x, -k_y, k_z)$ (refs. 38,47). The projection of the net OAM along the $y$-axis is zero if a $\mathcal{M}^*$ exists along the $xz$-plane. $\mathcal{M}^*$ can selectively exist for only one of the Fermi pockets at Γ or R depending on the crystallographic direction of applied current. For example, when the current passes along [100] ($z$-axis) and the NLH-induced chiral current is collected along [010] ($x$-axis), $\mathcal{M}^*$ exists along the $xz$-plane for the Fermi pocket at Γ, whereas it is broken for the Fermi pocket at R. Thus, the OAM contribution from topological bands at R would predominantly exist along the $y$-axis. Extended Data Fig. 2a(i),(ii) shows the third-order and second-order responses in each arm in a device fabricated in the mentioned crystallographic directions. We can observe that the chiral current is primarily generated in the left arm of the device. It represents the valve in the '$I^R$ on' state because the chiral current in the left arm preferentially exists in the Fermi pocket at R. The opposite is true when current is passed along the [100] ($z$-axis) and chiral current is collected along the [011] ($x$-axis). In this case, the OAM contribution from the topological bands at Γ would predominantly exist. Extended Data Fig. 2b(i),(ii) shows the nonlinear responses measured in a device made along these crystal directions. In this case, nonlinear currents are predominantly generated in the right arm of the device, thus representing the valve in the '$I^\Gamma$ on' position. Finally, Extended Data Fig. 2c(i),(ii) shows the nonlinear responses in a device in the 'valve off' states where the generation of chiral current is suppressed in both arms. In this case, the current is passed along [011] and NLH currents were collected along [100]. We call this strategy to control the valve position 'passive' because the valve tunability is linked with the intrinsic quantum geometry of the topological bands rather than the experimentally controlled parameter.

We now discuss a proof-of-concept study to achieve active tunability of the valve. As discussed in the main text, the chiral fermionic current

exists due to the preferential occupation of the R Fermi pocket in the left arm and Γ Fermi pocket in the right arm of the device. Thus, the magnitude of chiral currents can be controlled by tuning the occupational imbalance between two Fermi pockets. We fabricated a magnetic tunnel junction (MTJ) with in-plane magnetization on top of individual arms to locally probe the influence of the magnetic field on chiral currents. We developed a new fabrication technique to ensure the surface of the lamella is at a similar height to the substrate. Our strategy prevents any sudden height changes to ensure smooth deposition of the MTJ. Extended Data Fig. 3a shows the false-coloured SEM image of the prepared device with MTJ electrodes deposited on top of the left and right arms. The lamella was prepared in the 'valve on' position. In the first set of experiments, we studied the effects of the magnetization direction of the MTJ on the chiral current of both arms. $M_x$ and $M_z$ represent directions when the magnetization of MTJ is perpendicular and parallel to the current-induced orbital magnetization. Extended Data Fig. 3b(i),(ii) show the $V_{3\omega}$ responses at 77.77 Hz in the left and right arms, respectively, with $M_x$ and $M_z$ magnetizations. We can observe from Extended Data Fig. 3b that the chiral current has the same order of magnitude in the left and right arms for the $M_x$ orientation. However, the nonlinear response of the right arm switches sign when the MTJ magnetization was switched from $M_x$ to $M_z$; meanwhile, the $V_{3\omega}$ response in the left arm remains similar. Notably, these measurements were performed in the absence of an external magnetic field. The magnetic field was only used to switch the magnetization direction of the MTJ. Extended Data Fig. 3b shows a possible strategy to tune the valve state from 'valve on' to '$I^R$ on', based on a MTJ magnetization switching mechanism. In the second experiment, we varied the frequency of the applied current to study the interaction of the chiral current with the magnetization of MTJ by measuring the inductive impedance. Extended Data Fig. 3c shows the dependence of $V_{3\omega}$ responses at different frequencies of the applied current with the MTJ in the $M_x$ configuration. We note from Extended Data Fig. 3b(ii),c(ii) that the impedance response of $I^\Gamma$ is not significantly changed when the frequency is varied between 77.77 Hz and 47 Hz, whereas the impedance response of $I^R$ is reduced by half. From Extended Data Fig. 3c(i), we see that the magnitude of $I^R$ is further reduced by an order of magnitude upon going to the frequency of 23 Hz. Meanwhile, the $I^\Gamma$ response, although diminished, has the same sign and the order of magnitude. Through Extended Data Fig. 3c, we show that the valve can be tuned from 'valve on' position to '$I^\Gamma$ on' position by changing the frequency of the applied current from 77.77 Hz to 23 Hz with the MTJ in the $M_x$ configuration. Through these two experiments, we provide a first step to pursue active tunability of the chiral fermionic valve by its integration with an MTJ. However, a deeper understanding of the interaction between the chiral current and the magnetization dynamics is needed to further analyse these results. Also, a systematic study is crucial to analyse the switching reproducibility, fidelity and scalability of the MTJ-integrated device, which presents an important direction for future work.

## Quantum Interference below threshold current

We have shown the appearance of the $V_{3\omega}$ responses after a certain current threshold in Fig. 2. Meanwhile, Fig. 5b shows clear oscillations of $V_{3\omega}$ with applied current even below the threshold current. As shown in Extended Data Fig. 4a, we could also observe weak $V_{3\omega}$ oscillation response below the threshold current in the three-arm geometry as well. The subsequent question is that why does the interference occur even below the threshold current. We can explain the observed phenomenon using the schematic shown in Extended Data Fig. 4b. We discussed in the main text that the fermions in different Fermi pockets gain a transverse velocity due to the presence of $\Delta\boldsymbol{\Omega}_\varepsilon^{\Gamma,R}$. Equation (1) shows the magnitude of the current going into different arms of the device from different Fermi pockets is proportional to $\Delta\boldsymbol{\Omega}_\varepsilon^{\Gamma,R}$. The appearance of a nonlinear response after the current threshold suggests that $\Delta\boldsymbol{\Omega}_\varepsilon^{\Gamma,R}$ is non-zero only above it. However, $\boldsymbol{\Omega}_\varepsilon^{\Gamma,R}$ does exist on individual topological Fermi pockets even when $\Delta\boldsymbol{\Omega}_\varepsilon^{\Gamma,R}$ is zero. The

presence of $\boldsymbol{\Omega}_\varepsilon^{\Gamma,R}$ causes the fermions to contribute to the scattering equally in both arms of the device from both of the Fermi pockets. Therefore, the chiral current in each of the arms would not be preferentially carried by one of the Fermi pockets. However, the chiral currents in both of the arms do exist. But, the nonlinear responses of current with opposite chirality cancel out below the threshold current. $\Delta\boldsymbol{\Omega}_\varepsilon^{\Gamma,R}$ becomes non-zero above the threshold current. Thereby, the fermions from the Fermi pockets at Γ and R, are preferentially scattered into the right and left arms, respectively. This creates an occupational imbalance in these arms, which leads to the observation of the chiral current response from the individual Fermi pockets. The presence of chiral current in both scenarios makes it possible to observe the quantum interference of the chiral current even below the threshold current.

## Theoretical considerations for the NLH effect

We use the semi-classical Boltzmann formalism to derive the nonlinear responses. However, it is crucial first to discuss the limitations of the assumptions taken to derive these transport equations. The semi-classical Boltzmann approach considers electrons as an adiabatic Bloch wavepacket moving in a static band structure. We have shown with Figs. 2 and 3 that the nonlinear transport responses appear only above a certain current threshold. Therefore, the conventional Boltzmann approach cannot capture the electric field-induced transitions due to the non-equilibrium bands associated metric. Second, the formalism assumes a single distribution function per band, effectively treating all carriers as coming from a single Fermi surface. However, we have shown chirality-selective transport due to the imbalance in the occupation of two different Fermi pockets with opposite Chern numbers. These currents of different chirality may have different scattering and relaxation dynamics, which must be incorporated in the transport equations. Moreover, the assumption of a localized and non-interacting Bloch wavepacket used in the Boltzmann formalism cannot be used to accurately describe the phase-coherent transport of multifold fermions. Nevertheless, we have found that the Boltzmann formalism is very useful to qualitatively describe our experimental results.

The current density $\mathbf{J}_x$ flowing into the outer arms of the device along $x$, as shown in the schematic of Fig. 1c, is given by

$$\mathbf{J}_x = -e \int_{\mathbf{k}} f D \mathbf{v}_x \tag{2}$$

with $\int_{\mathbf{k}} x \equiv \int \frac{\mathrm{d}^3\mathbf{k}}{(2\pi)^3}$ and $f$ is the non-equilibrium distribution function, $\mathbf{v}_x$ is the velocity of an electron wavepacket along $x$ and $D$ is the modified density of states, given as

$$D = 1 + \frac{e}{\hbar}\mathbf{B}\cdot\boldsymbol{\Omega} \tag{3}$$

In the case of $B = 0$, the motion of electron wavepacket is given by the equation

$$\mathbf{v}_x = \frac{1}{\hbar}\frac{\partial\epsilon(k)}{\partial\mathbf{k}} + \frac{e}{\hbar}\boldsymbol{\mathcal{E}}\times\boldsymbol{\Omega} \tag{4}$$

where $\epsilon(k)$ is the energy dispersion relation. The first term is the group velocity, whereas the second term is the anomalous velocity. As described in the main text, we purposefully applied $\mathcal{E}_z$ along the principal axes of PdGa to minimize the Ohmic contribution of $\mathcal{E}_z$ along $x$. Thus, the group velocity along $x$ due to $\mathcal{E}_z$ would be close to zero. Thus incorporating $\mathbf{v}_x$ in equation (2) gives

$$\mathbf{J}_x = -\frac{e^2}{\hbar}\int_{\mathbf{k}} f\left(\mathcal{E}_z\boldsymbol{\Omega}_y\right) \tag{5}$$

In the relaxation time approximation, the non-equilibrium distribution function with a first-order correction is given by

$$f = f^0 + f^{(1)} = f^0 + \frac{e\tau\mathcal{E}_z}{1-i\omega\tau}\frac{\partial f^0}{\partial \mathbf{k}} \tag{6}$$

where $f^0$ is the Fermi–Dirac equation, $\tau$ is the intranode scattering and $\omega$ is the frequency of the applied a.c. electrical signal $\mathcal{E}_z(= E_0\sin(\omega t))$. On substituting equation (6) in equation (5), the current density $\mathbf{J}_x$ can be written as the sum of first-order and second-order electric-field terms $\mathbf{J}_x^{(1)}$ and $\mathbf{J}_x^{(2)}$ as

$$\mathbf{J}_x^{(1)} = -\frac{e^2}{\hbar}\int_{\mathbf{k}} f^0\,\mathbf{\Omega}_y\mathcal{E}_z \tag{7}$$

$$\mathbf{J}_x^{(2)} = -\frac{\tau e^3}{\hbar(1-i\omega\tau)}\int_{\mathbf{k}}\frac{\partial f^0}{\partial\mathbf{k}}\mathbf{\Omega}_y\mathcal{E}_z^2 \propto \int_{\mathbf{k}} f^0\,(\partial_{\mathbf{k}}\mathbf{\Omega}_y)\mathcal{E}_z^2 \tag{8}$$

In equation (7), the overall integration of $\mathbf{\Omega}_y$ throughout the Brillouin zone under TRS should be zero. However, $\mathbf{J}_x^{(1)}$ may contribute to first-order transverse current in our device geometry as shown in Fig. 1c under non-equilibrium steady-state conditions. However, there would be equal contribution from both of the Fermi pockets without any preferential scattering. The voltage response of $\mathbf{J}_x^{(1)}$ would be in-phase with the applied current, represented as $V_\omega^X$ in Extended Data Fig. 5(i). Meanwhile, $\mathbf{J}_x^{(2)}$ originating due to OAM dipole[37] would be of second-order, since $E_0^2\sin^2(\omega t) = E_0^2(1-\cos(2\omega t))/2$. Its voltage response would be $\pi/2$ shifted with respect to the applied current, represented as $V_{2\omega}^Y$ in Extended Data Fig. 5(ii). Extended Data Fig. 5a,b shows the experimental measurement of these responses for the left and right arms of the device.

In our experiments, the above equations correspond to our observed voltage responses below 55 μA, where the **g** is zero and the Berry curvature is the dominant quantum geometry component. As discussed in the main text, the quantum metric is non-zero above 55 μA and we get a field-induced Berry curvature $\mathbf{\Omega}_{\varepsilon,y} = \nabla_k \times (\mathbf{G}\mathcal{E}_z)$. Substituting this into $\mathbf{J}_x^{(1)}$ and $\mathbf{J}_x^{(2)}$ converts them into second-order and third-order transverse currents, respectively, which are given by

$$\mathbf{J}_x^{(2)} \propto \int_{\mathbf{k}} f^0\,(\nabla_{\mathbf{k}}\times(\mathbf{G}\mathcal{E}_z))\mathcal{E}_z \tag{9}$$

$$\mathbf{J}_x^{(3)} \propto \int_{\mathbf{k}} f^0\,\partial_{\mathbf{k}}(\nabla_{\mathbf{k}}\times(\mathbf{G}\mathcal{E}_z))\mathcal{E}_z^2 \tag{10}$$

These equations are the basis of the experimental data discussed in the main text in Fig. 2 and Fig. 3. $\mathbf{J}_x^{(3)}$ will contribute to a first-order and third-order current response, because $E_0^3\sin^3\omega t = E_0^3\left(\frac{3}{4}\sin\omega t + \frac{1}{4}\sin(3\omega t + \pi)\right)$. Their voltage response would be in-phase with the applied current applied current, represented as $V_\omega^X$ and $V_{3\omega}^X$ in Extended Data Fig. 5.

We also observed $\pi/2$ shifted responses of $V_\omega^X$, $V_{2\omega}^Y$ and $V_{3\omega}^X$. These responses are shown in Extended Data Fig. 6, Fig. 3 and Fig. 2(i), respectively. The appearance of $V_\omega^Y$, $V_{2\omega}^X$ and $V_{3\omega}^Y$ due to $\mathbf{J}_x^{(1)}$, $\mathbf{J}_x^{(2)}$ and $\mathbf{J}_x^{(3)}$, respectively suggests the presence of an inductive impedance, as suggested by Faraday's law. These responses only start to dominate after the current threshold of 55 μA is reached when **g** is non-zero. It suggests that the inductance experienced by the chiral fermionic current in the right and left arms is the function of $\mathbf{g}^\Gamma$ and $\mathbf{g}^R$, respectively. The dependence of $V_\omega^Y$, $V_{2\omega}^X$ and $V_{3\omega}^Y$ responses on **g** allowed us to capture its opposite sign for topological Fermi pockets at $\Gamma$ and R points. Notably, it also suggests that the chiral fermionic current carries orbital angular moments. As in our case, it will lead to generation of a spin current since PdGa has a large spin–orbit coupling[3]. The inductance values for the chiral currents following into the right and left arm can convey the magnitude of these currents. We used the overall applied current applied in the device in our calculation, which gives the lower limit estimation of the inductance reactance, as given in Extended Data Fig. 6.

## Theoretical considerations for the quantum interference of chiral current

The phase acquired by a free electron due to electric-field-induced Berry connection ($\mathbf{A}_\varepsilon$) in the absence of an external magnetic field along the path $j$ is given by

$$\theta = \frac{e}{\hbar}\int \mathbf{A}_\varepsilon \cdot \mathrm{d}\mathbf{l}_j \tag{11}$$

In our system the vector potential $\mathbf{A}_\varepsilon$ experienced by an electron in the left and right arms is $\mathbf{G}^R\mathcal{E}_z$ and $\mathbf{G}^\Gamma\mathcal{E}_z$, respectively. Thus, the phase difference due to the acquired by the electron travelling along the left ($\theta_R$) and right ($\theta_\Gamma$) arms is given by

$$\Delta\theta = \theta_R - \theta_\Gamma = \frac{e}{\hbar}\left(\int \mathbf{G}^R\mathcal{E}_z \cdot \mathrm{d}\mathbf{l}_j - \int \mathbf{G}^\Gamma\mathcal{E}_z \cdot \mathrm{d}\mathbf{l}_j\right) \tag{12}$$

Assuming **G** does not vary spatially, we can rewrite the above equation as

$$\Delta\theta = \frac{e}{\hbar}(\mathbf{G}^R - \mathbf{G}^\Gamma)\Delta V_z \tag{13}$$

where $\Delta V_z$ is the voltage difference across the MZI. The term $\mathbf{G}^R - \mathbf{G}^\Gamma$ is non-zero as **G** of Fermi pockets at R and $\Gamma$ have opposite signs. Thus, we would observe the oscillation in phase on applied current, as shown in Fig. 5b.

## Temperature dependence

Extended Data Fig. 7a-c show the log-scaled temperature dependence of the first-order, $V_{3\omega}$ and $V_{2\omega}$ longitudinal responses in one of the arms of the device. Extended Data Fig. 7a shows that the first-order response decreases with temperature till 15 K after which it starts to saturate. Upon cooling further, the $V_{3\omega}$ and $V_{2\omega}$ responses start to appear, as shown in Extended Data Fig. 7b,c. The appearance of the $V_{3\omega}$ response is concurrent with the observed upturn in the first-order response, as expected from equation (1). The $V_{3\omega}$ and $V_{2\omega}$ responses start to appear below 3.4 K and 14 K, respectively. The exact transition temperature varied between devices, with different device geometries and crystallographic orientations. However, the concurrent appearance in the upturn of the first and third-order responses was always observed in all the measured devices. Data at lower temperatures were measured in a Bluefors system which showed that the nonlinear responses start to saturate below 1 K. Note that these data are not shown because they had a considerably better signal-to-noise ratio due to the low-noise filters present in the Bluefors system. Extended Data Fig. 7d,e shows the relative modulation of the first-order responses with magnetic field orientation in the right and left arms, respectively, at different temperatures. The modulation in the different arms remains of opposite signs below 100 K even in the device with 'valve off' position. After 100 K, the change in the sign of the modulation in the right arm was observed. Thereafter, both arms share the same sign of modulation, albeit of different magnitudes. Above 200 K, the modulation with magnetic field orientation starts to disappear in both of these arms.

Extended Data Fig. 7f shows the temperature dependence of the FFT amplitude of the $V_{3\omega}$ oscillation with the magnetic field. The amplitude of the oscillation decays slowly with temperature below 1 K, after which it starts to decreases more sharply with temperature. The amplitude decay trend matches the temperature dependence of the magnitude of the $V_{2\omega}$ shown in Extended Data Fig. 7c. We strongly believe that the FFT amplitude is linked with the quantum metric magnitude. The $V_{2\omega}$ response is the direct indicator of quantum metric response. Assuming amplitude decay solely due to an inelastic mechanism would not be accurate as the quantum metric does not scale with the scattering time in Drude conductivity. Therefore, the amplitude decay was not

correlated to the phase-coherent length because the mechanism of decoherence of chiral fermions is not completely known. We also tried to measure the amplitude of the oscillation of the $V_{3\omega}$ response with applied current shown in Fig. 5b. We observed that the oscillation period changed with temperature, and the FFT amplitude broadened into multiple peaks. Hence, tracking the FFT amplitude with temperature was not trivial. It may be due to temperature variation of current-induced magnetization, which influences the oscillation period.

## Possible trivial-state contributions to current directionality

PdGa belongs to a gyrotropic class with the tetrahedral chiral point group 23 (T), which allows for a Dresselhaus-type spin–orbit coupling[48]. The spin–orbit coupling can cause splitting of the trivial band (and topological bands). The applied electric field creates non-equilibrium spin polarization of both spins, as shown in Extended Data Fig. 8a. The current applied along $z$ would create a spin polarization for both spins along $z$. A previous study in 2002 proposed the spin galvanic effect, in which electric current was produced due to spatially uniform non-equilibrium spin polarization[49]. Empirically, the electric current density ($j_\alpha$) is linked with the average spin of the electron ($S_\gamma$) by $j_\alpha = \sum Q_{\alpha\gamma}S_\gamma$, where $Q$ is the second-rank pseudotensor of a gyrotropic crystal. These currents are semi-classically modelled using spin-dependent scattering asymmetry. In the conventional spin galvanic effect, the current is galvanized because of the asymmetry in spin-flip scattering, which requires a spin population imbalance of a particular spin. However, opposite spin currents can also be galvanized because of the asymmetry in skew scattering, as in the case of inverse spin Hall effect, as schematically shown in Extended Data Fig. 8b. Similar to the spin galvanic effect, the inverse spin Hall effect also generates a dipolar term proportional to ($k \cdot S$) because of spin-dependent correction of the Fermi–Dirac distribution ($\delta f_{k,S}$) (ref. 50). For non-zero $Q_{xz}$, this would generate currents of opposite spins in different arms of the device, instead of the spin Hall voltage measured in a conventional Hall device. The spin currents generated due to inverse spin Hall effect (or spin galvanic effect) can give the desired current directionally solely due to the trivial bands of PdGa. This filtration of spin current into different arms is conceptually similar to the filtration of the chiral fermions due to electric field-induced quantum geometry.

The magnitude of opposite spin currents galvanized into the outer arms would depend on the size of the Fermi surface[51] and $\delta f_{k,S}$. These parameters are similar for both spin currents originating from the Fermi surface of a same trivial band. Therefore, we would expect that the relative magnitude of current in both arms would remain similar. It would be irrespective of the crystallographic direction of applied current, even when the magnitude of individual spin current may vary because of anisotropic skew scattering. In Extended Data Fig. 2, we have shown that we can tune the relative magnitude of the generated current in both arms by passing current in different crystal directions. Therefore, the spin currents galvanized due to $Q_{\alpha\gamma}$ of trivial bands cannot explain the differences in the relative magnitude of currents measured in different devices. We will now provide the second evidence by ruling out the involvement of the $Q_{\alpha\gamma}$-like tensor from the trivial bands. We measured anomalous Hall response in three devices with crystallographic orientation corresponding to 'valve on', '$I^R$ on' and '$I^\Gamma$ on' positions. Extended Data Fig. 8c shows the electrical configuration used to measure the Hall responses. The magnetic field was rotated in the $yz$ plane, where $\theta = 90°$ corresponds to the field along the direction of applied current $z$. Extended Data Fig. 8d shows the relative change in the magnitude of the Hall response on magnetic orientation in a device with 'valve on' position with respect to $\theta = 90°$. The response in this valve position resembles the response expected from a trivial Hall effect. $\Delta\boldsymbol{\Omega}_\varepsilon^\Gamma$ and $\Delta\boldsymbol{\Omega}_\varepsilon^R$ have a similar magnitude contribution to the OAM dipole in the 'valve on' position. Therefore, there is no anomalous Hall response due to the absence of any net

magnetization. The Hall response was also similar in the device in 'valve off' state, which is expected because of the absence of the OAM dipole itself along the $y$-axis. Extended Data Fig. 8e,f show the relative change in the magnitude of the Hall response with the magnetic field orientation in the device in '$I^R$ on' and '$I^\Gamma$ on' position, respectively. These responses show the presence of an anomalous Hall response with distinct symmetries. In '$I^R$ on' position, a net magnetization is present because $\Delta\boldsymbol{\Omega}_\varepsilon^R$ has larger contribution in the OAM dipole than $\Delta\boldsymbol{\Omega}_\varepsilon^\Gamma$. The symmetry of the Hall responses matches the symmetry of the longitudinal response of chiral current discussed in Fig. 4b(i). Meanwhile, $\Delta\boldsymbol{\Omega}_\varepsilon^\Gamma$ contributes more to the OAM dipole in '$I^\Gamma$ on' position; therefore, the anomalous Hall response captures its two-fold symmetry. The three distinct anomalous Hall responses observed in our study cannot be explained by $Q_{\alpha\gamma}$-driven currents. Our results indicate that the observed two distinct Hall responses can be explained only by considering the $\Delta\boldsymbol{\Omega}_\varepsilon^\Gamma$ and $\Delta\boldsymbol{\Omega}_\varepsilon^R$ contributions coming from two different topological Fermi pockets.

## Role of Fermi-arcs in long-range coherence

The phase coherence of electrons in a typical metal such as Cu is typically of the order of few tens of nanometre. However, we show in our device that the phase coherence length of chiral current is above 15 μm. The long coherence length is due to the chiral nature of the charge carriers in the topological states. There are three possibilities of electronic states for fermions to occupy when they scatter into the left or right arm of the device due to $v_a$. They can scatter into (1) trivial states; (2) topological states with opposite chirality; or (3) topological states with the same chirality. The chiral fermions scattering into trivial states would violate the Nielsen–Ninomiya theorem[23]. It would imply fermions losing their chirality on scattering into non-chiral trivial states. Thus, the decoherence of chiral current into trivial states cannot occur even in the presence of empty trivial states near the Fermi-level. The fermions can also undergo inter-valley scattering into the topological states with opposite Chern number. Notably, Fermi-arc states existing on the surface provide a direct pathway for the fermions to switch their chirality as it connects the topological band crossings with opposite Chern numbers. We present our hypothesis of the role of Fermi-arcs in preserving the phase coherence of chiral fermionic currents. There are two pairs of spin-split Fermi-arcs that connect the Γ point to the R points at the corners of the Brillion zone at the top and bottom surfaces[3]. The chiral current $I_\omega^R$ can leak into the Γ band by the Femi-arcs present on the surface, whereas the opposite will occur for the chiral current $I_\omega^\Gamma$. Thus, the flow of the leakage current is outward from Γ to R in the right arm, whereas it is inwards towards Γ from R in the left arm. We posit that the presence of current-induced magnetization of opposite polarities prevents the charge relaxation through Fermi-arcs in both arms. Consequently, the chiral current due to the preferential occupancy of topological bands can have longer relaxation times.

## Data availability

The datasets used in this study to generate the main text figures are available at Zenodo[52] (https://doi.org/10.5281/zenodo.15190411). Additional data are available upon request.

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

**Acknowledgements** We thank A. K. Dagnino and T. Neupert for their help with the theoretical analysis presented in the Supplementary Information. We thank N. Schröter, B. Pal and A. Johannsen for their discussions.

**Author contributions** A.D., C.F. and S.S.P.P. conceived the project. A.D. conceptualized the device geometry, fabricated the FIB microstructured device, and conducted the experiments and data analysis. K.M. synthesized and characterized the crystals. A.D. performed the theoretical analysis to qualitatively describe the experimental data. P.K.S. assisted with the initial transport measurements. A.D. and S.S.P.P. wrote the manuscript, with input from all authors.

**Funding** Open access funding provided by Max Planck Society.

**Competing interests** The authors declare no competing interests.

**Additional information**
**Correspondence and requests for materials** should be addressed to Claudia Felser or Stuart S. P. Parkin.

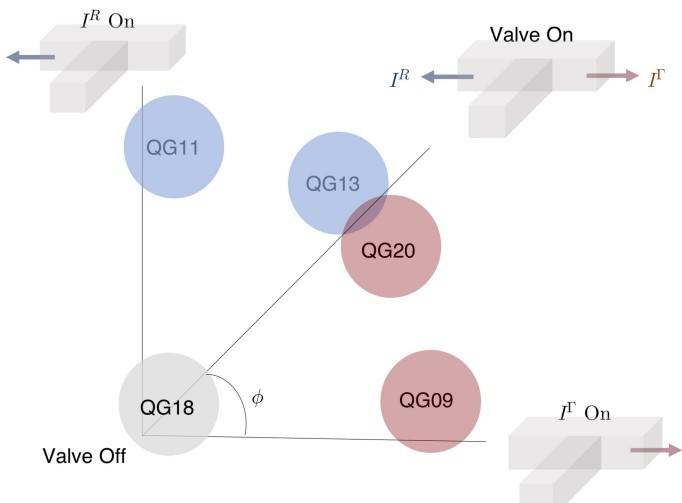

**Extended Data Fig. 1 | Quantification of Valve positions.** Schematic showing the devices mentioned in Extended Data Table 1, categorised into different valve positions. The valve positions are quantified using the parameter φ.

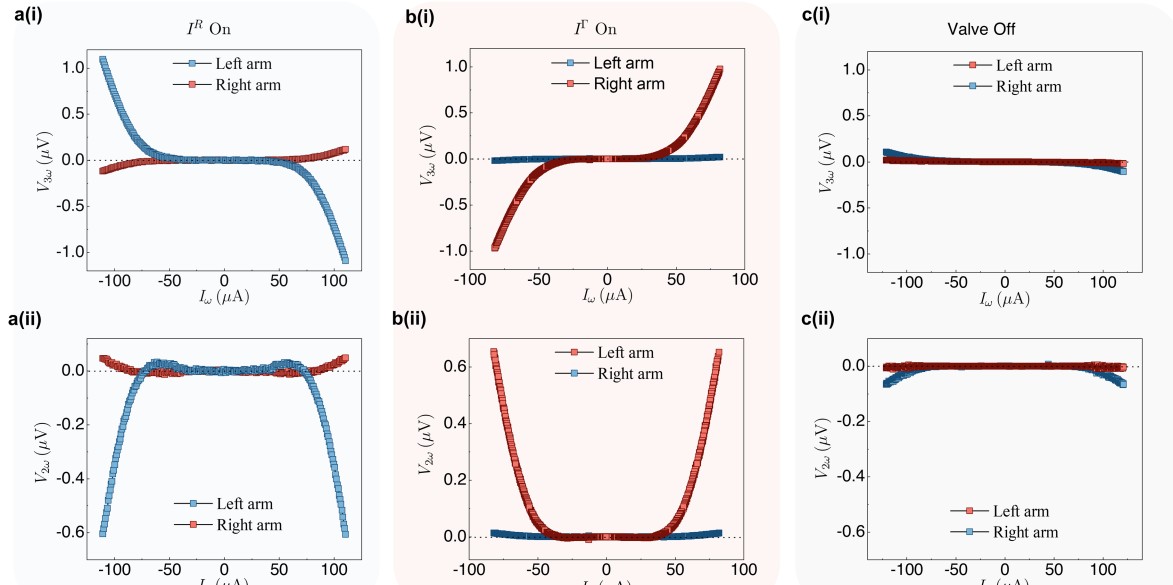

**Extended Data Fig. 2 | Passive control of the chiral fermionic valve.**
**a–c** Comparison of the third-order (a(i)-c(i)) and second-order (a(ii)-c(ii)) responses in the left and right arms of the device in the '$I^R$ On' (left column), '$I^\Gamma$ On' (middle column) and 'Valve Off' (right column) positions. The electrical configuration used to measure these responses was the same as shown in Fig. 2.

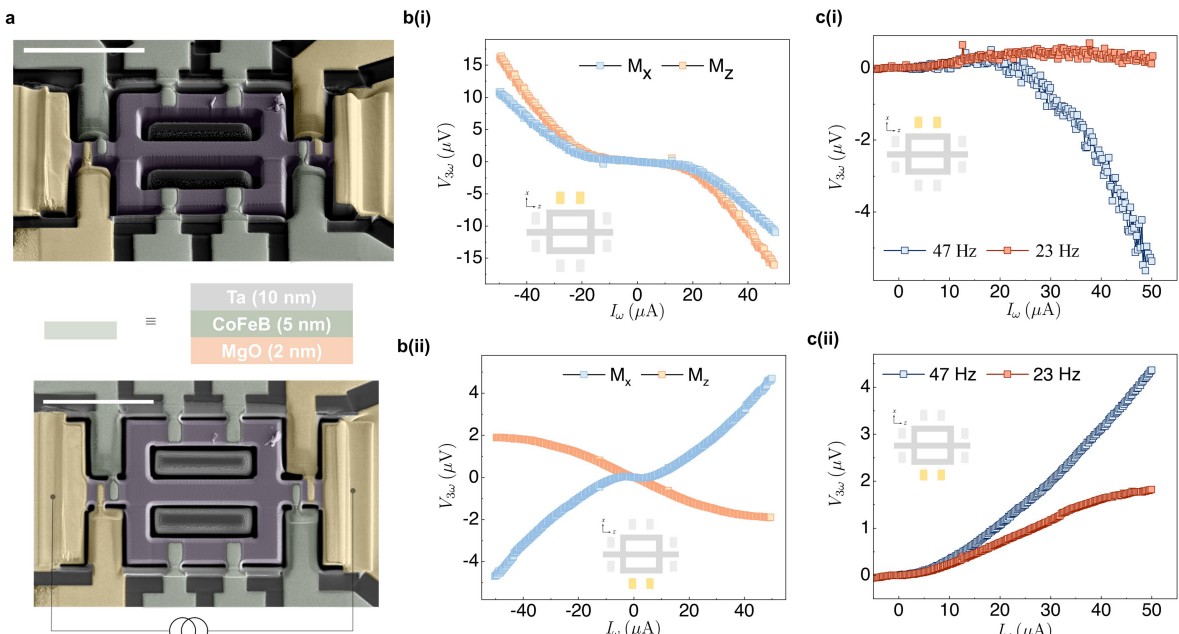

**Extended Data Fig. 3 | Active control of the chiral fermionic valve. (a)** (top) False-coloured SEM image of the device prepared by placing the lamella inside the trench made in the substrate. (middle) Schematic of the MTJ stack deposited on top of the device. (bottom) Top view of the device. The scale bar is 10 μm. **b(i)-(ii)** Dependence of the third-order response in the left and right arms of the device in the MTJ configuration with magnetization direction along the X and Z directions at a 77.77 Hz frequency of the applied current. **c(i)-(ii)** The variation of the third-order response in left and right arm with the frequency of the applied current. The magnetization of the MTJ was along X. The electronic configuration was the same as shown in Fig. 4 of the main text.

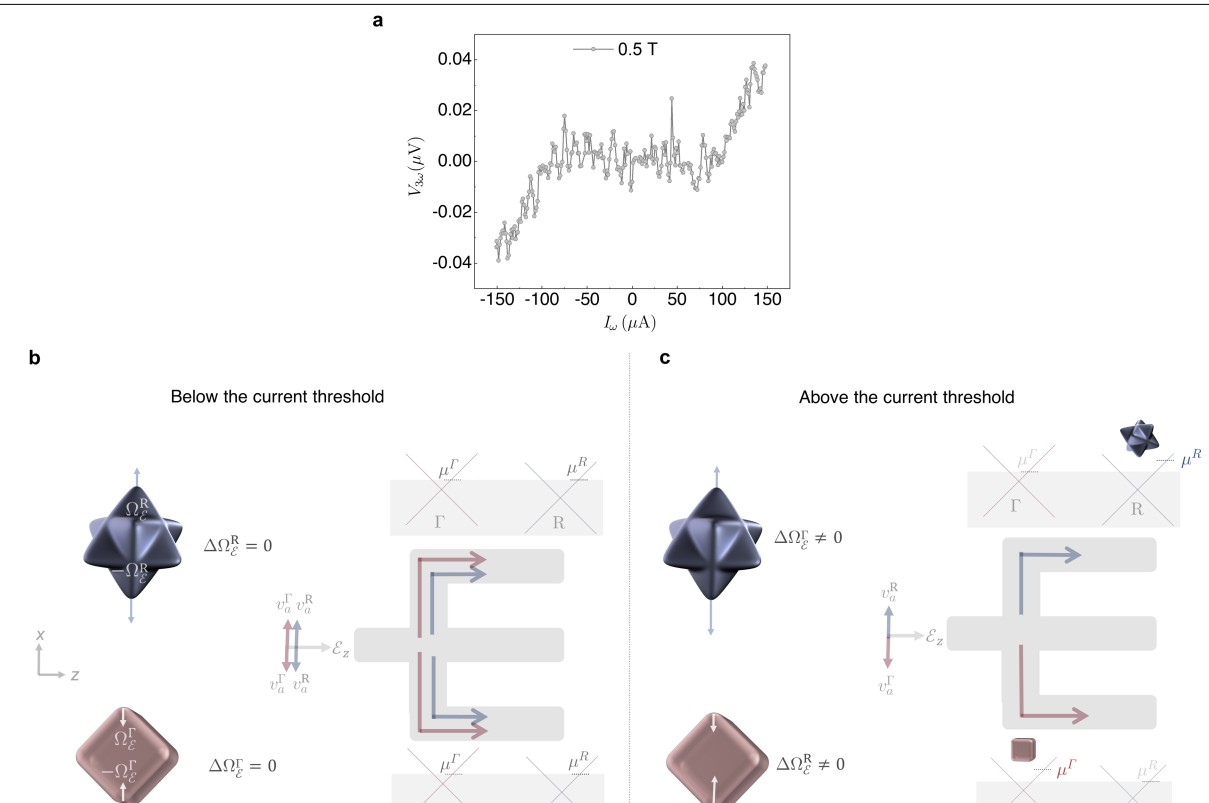

**Extended Data Fig. 4 | Interference below the threshold current. a**. Oscillations in the third-order response with applied current at 3.4 K in the three-arm geometry at zero external magnetic field. **b,c,** Schematic showing the generation of chiral currents from different Fermi-pockets below (**b**) and above (**c**) the threshold current.

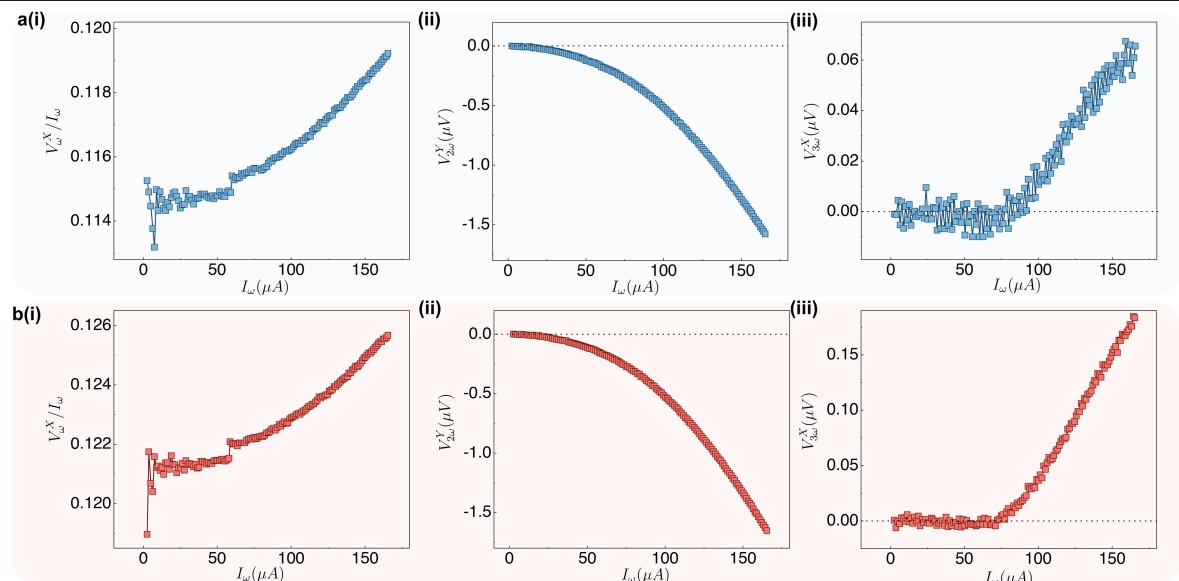

**Extended Data Fig. 5 | NLH responses as per Eq. 7–10.** The electrical configuration to measure these responses was same as in Fig. 2. **a(i)-(iii)** show the dependence of first-order $\frac{V_\omega^X}{I_\omega}$, second-order $V_{2\omega}^Y$ and third-order $V_{3\omega}^X$ responses on the applied current $I_\omega$ at 3.5 K for the currents $I_\omega^R$, $I_{2\omega}^R$ and $I_{3\omega}^R$ entering the left arm of the device, respectively. **b(i)-(iii)** show these responses for the currents $I_\omega^\Gamma$, $I_{2\omega}^\Gamma$ and $I_{3\omega}^\Gamma$ going into the right arm, respectively.

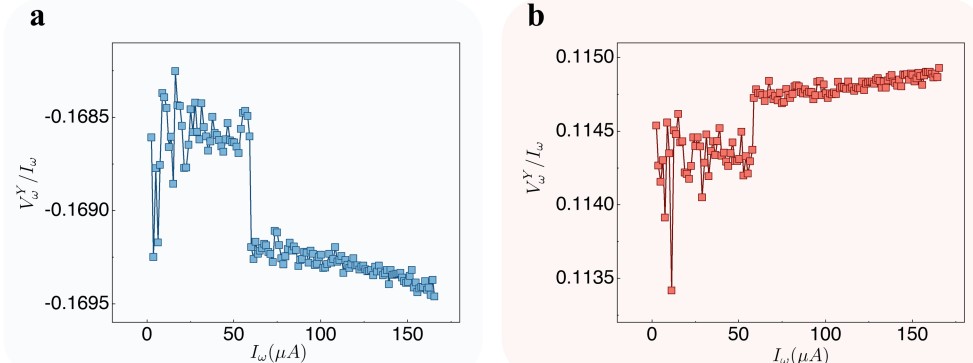

**Extended Data Fig. 6 | Inductive reactance of chiral current. a-b** Inductive reactance of chiral fermionic currents flowing into the left and right arms at 3.5 K respectively. The electrical configurations are the same as in Fig. 2.

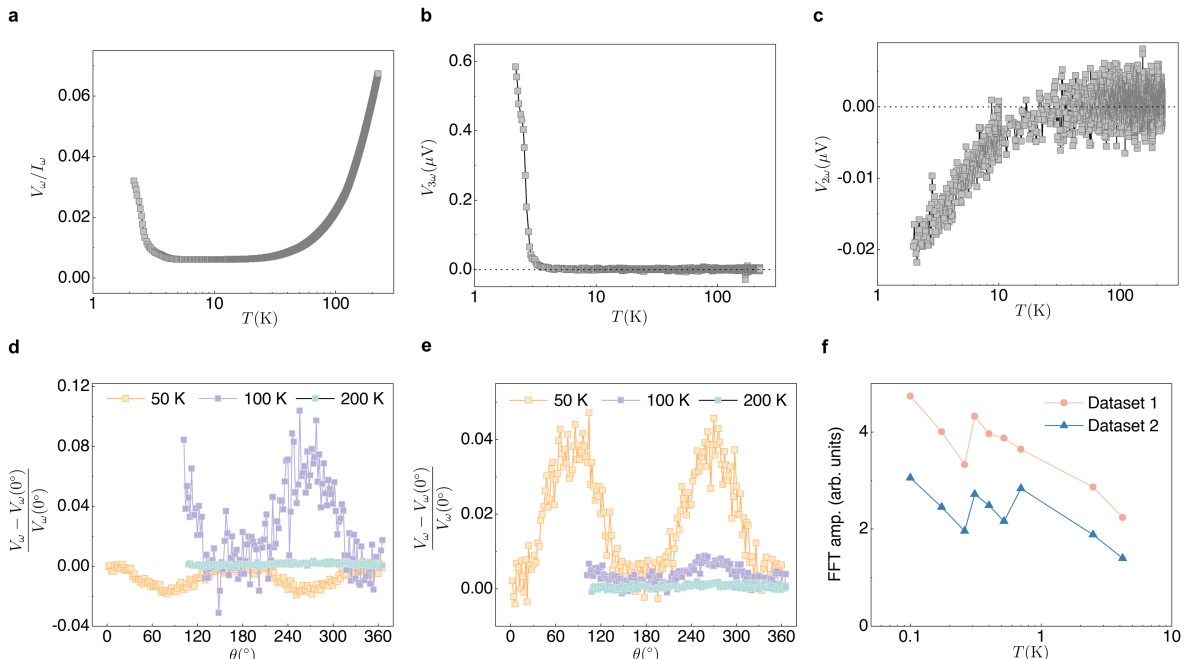

**Extended Data Fig. 7 | Temperature dependence of the NLH responses and quantum interference. a-c** Variation of the first, third, and second-order responses in one of the devices' arms. **d-e** Relative variation of the first-order response in the right and left arm of the device, respectively, with magnetic field orientation at different temperatures with an applied current of 400 μA.

The magnetic field of 2 T was rotated in the xy plane, as discussed in Fig. 4 of the manuscript. **f** Temperature dependence of the FFT amplitude of the magnetic fields induced oscillation of the third-order response. Note that these data, with the exception of the data in (f) were measured in a PPMS system (base temperature 2 K).

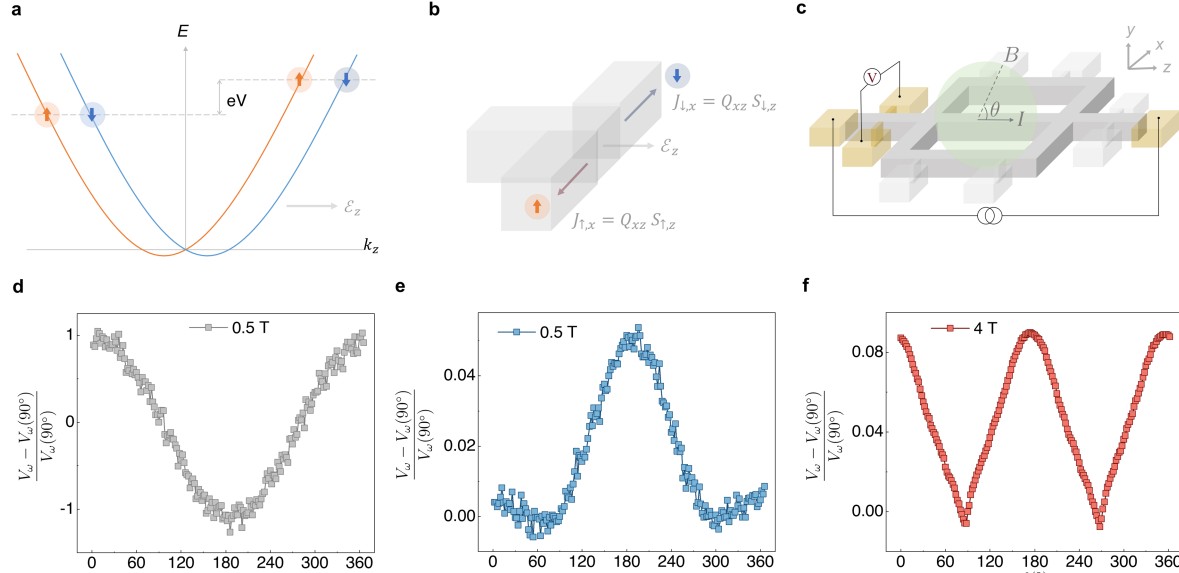

**Extended Data Fig. 8 | Anomalous Hall effect. a** The Rashba split trivial bands in the presence of spin-orbit coupling. **b** Spin current galvanised due to the presence of $Q_{\alpha\gamma}$ of the spin-split trivial band. **c** The electronic configuration used to measure the Hall responses. **d** The Hall response of the device in the 'Valve On' state. **e-f** The relative variation for the Hall response with the magnetic field orientation in the device with 'I$^R$ On' and 'I$^r$ On' positions. These responses were measured using 400 µA of applied current at 2 K.

**Extended Data Table 1 | Summary of different fabricated devices**

| Device name | z-direction | x-direction | $\phi = \tan^{-1}\dfrac{V_{3\omega}^{R}}{V_{3\omega}^{\Gamma}}$ |
|---|---|---|---|
| QG20 | $[0\bar{1}1]$ | [100] | 38.65° |
| QG13 | [100] | 5° away from $[0\bar{1}1]$ | 49.43° |
| QG09 | [100] | 8° away from [011] | 4.84° |
| QG18 | [011] | [100] | OFF |
| QG11 | [100] | 8° away from [010] | 83.88° |

**a** Table showing the crystal orientation of the devices fabricated in different valve positions. The current is applied along z, and the NLH currents are collected along x. The valve positions are quantified using the parameter Φ. The third-order voltages were measured at 80 μA at 3.4 K to calculate Φ. The crystal is well oriented along [100], therefore the error margin while fabricating devices is within 1–2 degrees. However, the crystal is not well aligned along other directions; thus, the margin for error is a bit higher, around 4–5 degrees.

**Extended Data Table 2 | Summary of devices with Φ close to 45°**

| Device name | Dimensions [l, w, t] in μm | $I_{threshold}$ in μA | $(V_{2\omega}, V_{3\omega})$ in μV |
|---|---|---|---|
| QG13 | [5.85, 1.4, 1.3] | 55 | (0.01, 0.036) |
| QG20 | [6.4, 2, 2.7] | 58.8 | (0.028, 0.04) |
| QG17 | [6, 2.6, 3.3] | 57.8 | (0.12, 0.28)* |
| QG21 | [24.3, 2.3, 3] | 57.9 | (0.01, 0.026) |

*at 50 mK

Table showing the threshold current and the magnitude of the NLH responses measured for devices having Φ closer to 45°. [l,w,t] corresponds to the length, width, and thickness of the conduction channel. The length was measured between the voltage probes of an individual arm. NLH responses were measured at 70 μA and 3.4 K except for the device labelled with * that was measured at 50 mK.