## [Peer Review file · Nature]

A Chiral Fermionic Valve driven by Quantum Geometry

Corresponding Author: Professor Stuart Parkin

Version 1:

Reviewer comments:

Referee #1

(Remarks to the Author)

This manuscript reports a novel mechanism for chirality-based filtering of fermionic currents using the quantum geometry of multifold topological semimetals, demonstrated in PdGa. The authors design a three-arm device that spatially separates topological chiral fermions without magnetic fields, detect their orbital magnetization, and demonstrate their phase coherence via Mach-Zehnder interferometry. The work introduces the concept of a “chiral fermionic valve”, highlighting the control and coherence of chiral topological transport.

The study is bold, imaginative, and potentially transformative for topological quantum electronics. The experimental design is clever, and the results suggest new device paradigms. However, the manuscript also has areas where claims are overstated, and further clarifications or evidence are needed.

The idea of using quantum geometric curvature (Berry curvature and quantum metric tensor) to achieve chirality filtering without magnetic fields is conceptually significant. In addition, the three-arm device and the Mach-Zehnder geometry are well-conceived and effectively probe chiral separation and coherence. It is also remarkable that the measurements across different harmonics (1ω , 2ω , 3ω) and their dependence on current and field orientation are thorough and observation of interference oscillations (Fig. 5) is impressive, suggesting long-range phase coherence in chiral topological states.

However, this manuscript is not free of major concerns.

The following are a few examples:

1. The authors attribute current directionality to topological states based on anomalous velocity, but it's not clearly disentangled from possible trivial contributions. Evidence ruling out trivial state effects is circumstantial.
2. The term "valve" implies a degree of tunability or switchability (e.g., as in a transistor or diode). While spatial separation is shown, active control (beyond changing current magnitude or direction) is limited. This weakens the analogy to classical “valves”.
3. Several theoretical derivations (e.g., Eq. 1 and subsequent semiclassical analysis) are presented without discussing limitations of approximations, especially in systems with spin-1 and spin-3/2 quasiparticles.
4. The discussion on Fermi arcs as a mechanism for chirality coherence and protection is speculative (pp. 20-21). This section should be better supported or more clearly marked as a hypothesis.
5. Only a few devices are presented. While it is stated that multiple devices were made, results shown focus on single examples. Statistical robustness is missing.
6. The effect of temperature on coherence length, scattering, and nonlinear responses is not explored in depth. This is critical to understand the practical viability of such devices.

A few minor concerns:

1. Several sentences are verbose and could benefit from tightening. For example, the abstract can be made crisper for clarity.
2. Repeated use of phrases like “remarkably” or “demonstrate” makes some claims feel overstated.
3. Several claims about chiral coherence, magnetization, and device implications would benefit from additional references, particularly to related work in non-linear Hall effects and topological electronics.

All in all, the manuscript has the potential to be a high-impact publication in Nature, but it currently overstates its conclusions and needs stronger evidence or clarification in key areas. The novelty of the chiral fermionic valve concept is substantial, but its practical tunability and control are still preliminary. A deeper discussion of limitations, more evidence excluding trivial contributions, and clearer exposition of theoretical assumptions will significantly improve the manuscript.

Referee #2

(Remarks to the Author)

In this manuscript, Dixit et al. fabricated the nanoscale Hall device from a single crystal of PdGa to separate the currents with opposite fermionic chiralities by exploiting the quantum geometry of topological bands. They then observed the quantum interference of these chiral currents with mesoscopic phase coherence. The demonstration of mesoscopic coherent transport of chiral fermions is crucial to realizing functional quantum electronics. Their strategy to separate the left- and right-handed chiral fermionic currents and to interfere with them is very attractive. These chiral fermionic devices pave the way for practical applications that harness the chiral degree of freedom in topological quantum materials. While their experimental results are very interesting, there are several aspects that need further clarification.

(1) As reported in this work, the nonlinear Hall (NLH) effect exhibits an obvious threshold current of around 55 μA . Above this threshold current, both the second and third NLH signals appear. They attribute this transition to the appearance of the NLH currents. However, it is not clear to me why the NLH currents can induce such a transition. Moreover, there is a jump in AC resistance at a similar current. The authors claim that this resistance jump was not observed in DC measurements. What is the resistance difference between AC and DC measurements?

(2) What is the relationship between the magnitude of NLH signals and applied currents? Conventionally, the relationship between $V_{3\omega}$ ($V_{2\omega}$) and $I\omega$ follows the cubic (square) law.

(3) The measured current is the total current that flows through all left, right and middle arms. Does the ratio of the currents flowing in the three arms remain the same at different currents?

(4) It is noticed that in Fig. 4b(ii), the magnetic field and device temperature are significantly different from those in other panels. And there are significant differences in signal magnitude. Is there an explanation for its cause?

(5) In the interference measurement, they removed the middle arm. Is it just to mimic the Mach-Zehnder interferometer? What is the effect of the middle arm on the interference? Without the middle arm, the trivial current will flow into the left and right arms, which still induces a background electrical signal.

(6) They observed the quantum interference of chiral current at the current range of $\pm 100 \mu\text{A}$ and field range of 0-1 T. Can larger currents/magnetic fields (negative field?) maintain coherence and interference? I also notice that the device temperature needs to be very low (30 mK) to observe the interference effect. Is this interference effect limited by the coherence length, which dramatically decreases with the temperature? If so, how is the relationship between the coherence length and the device temperature?

(7) This question is related to comment (1). Unlike the device with a middle arm (Fig. 2), there is no transition of NLH signals with respect to current in the interference device (Fig. 5b). The device can work at a relatively low current of 20 μA . What is the difference between the two devices? Temperature or device geometry?

(8) Is there any detailed explanation about how the chiral currents acquire a phase due to the current-induced orbital magnetizations? I strongly recommend that the authors add a schematic to illustrate the mechanism of the current- and field-controlled quantum interference.

(9) This question is about the reliability of the experimental results. How many PdGa devices have they fabricated and measured? Do they exhibit similar performances in terms of threshold currents and the magnitude of NLH signals? Can they fabricate several quantum interference devices with different arm lengths? If so, they can quantify the coherence length, which is a very important parameter for practical applications.

Referee #3

(Remarks to the Author)

This manuscript reports a study on chiral fermionic transport in a three-arm device fabricated from the chiral topological semimetal PdGa. The authors fabricate high-quality device using advanced microfabrication techniques (focused-ion-beam milling). Through systematic investigations of second/third-order nonlinear Hall responses and magnetic field dependencies, the authors demonstrate the spatial separation of chiral fermions with opposite chirality into distinct arms of the device. Additionally, they show that the quantum interference of the chiral currents can be observed by using a Mach-Zehnder interferometer, confirming their long-range phase coherence. This work provides novel insights into the design of topological electronic devices based on quantum geometry and holds significant scientific and practical potential. I recommend the manuscript for publication, subject to revisions listed below:

1. The semiclassical Boltzmann kinetics (Eqs. S1–S6) provides qualitative insights into the nonlinear responses but lacks validation against effective model or first-principles calculations. Incorporating numerical calculations of nonlinear conductivity based on Berry curvature and quantum metric would strengthen the theoretical framework.

2. There are several distinct mechanisms of nonlinear transport, each subject to different symmetric constraints and scaling behaviors. However, the authors arbitrarily attribute the observed (2nd and 3rd order) nonlinear signals to the quantum geometric quantities (g and Ω) without establishing a rigorous connection to crystal symmetry or band structure. The authors should provide further explanation.

3. The authors devoted considerable words explaining the Fermi surfaces and Chern numbers at different high-symmetry points. However, the lack of both intuitive band structure presentation and explicit Chern number calculations weakens their analysis. Further addition of the first-principles calculations would enhance the clarity of their discussion.

4. In lines 165-167, the authors declare that PdGa breaks P and T simultaneously, thus attribute the nonlinear Hall signal to the quantum metric g . Moreover, in lines 171-173 the authors claim that a nonzero g is the evidence of T breaking. This claim relies on the very assumption it is meant to prove. Considering PdGa is a non-magnetic material, the authors need to provide more explanations of this point.

5. The expressions of nonlinear transport [Eqs. (1), (S6a), and (S6b)] only consider the electric field modified Berry curvature, but the electric field modified band energy is ignored, which affect the Fermi-Dirac distribution f_0 . The absence of this modification gives totally conflict results, especially in symmetric constrains. The authors need carefully examine the relevant references [PRL 112,16601 (2014), PRL 127,277201 (2021), PRL 127,277202 (2021), NSR nwaee334 (2024)].

6. The authors emphasize the significance of the quantum metric in nonlinear transport throughout the manuscript. However, they have completely confused this concept—in the original paper [PRL 112,16601 (2014)], " g " represents the Berry curvature polarizability (BCP) or the band-normalized quantum metric, not the quantum metric. The authors are strongly advised to carefully reorganize the theoretical analysis.

7. In line 50, the authors refer to the quantum geometric tensor T and provides its expression. Most strikingly, the authors exclusively use tensor arrow notation exclusively for the quantum metric tensor, while completely omitting such notation for the quantum geometric tensor and the Berry curvature tensor. This misleading notation creates unnecessary confusion and raising serious concerns about this manuscript.

8. The manuscript contains numerous notational inconsistencies. For instance, the symbol $\Delta\Omega$ is ambiguously used to denote both field-modified Berry curvature and orbital angular momentum dipole. We strongly recommend the authors to rigorously clarify their notational scheme.

9. The manuscript contains numerous nonstandard and erroneous vector notations. According to conventional typographic practices, bold font should denote vectors/tensors while regular font with subscript should indicate components. However, the authors violate this convention (e.g., $v_a = eE \times \Omega$), with similar confusing notation appearing in at least 4 different places. We strongly recommend the authors conduct a thorough revision to rectify all issues throughout the text.

10. A series of typos need be fixed. Line 60: A period at the end of the sentence is missing. Line 439: f is not in the formular environment. Line 441: Redundant parentheses.

Version 2:

Reviewer comments:

Referee #1

(Remarks to the Author)

The revised manuscript includes new experimental data across multiple devices, detailed temperature dependence, additional anomalous Hall measurements, and a discussion of theoretical limitations. The concept of the chiral fermionic valve is now supported by both passive control achieved through crystallographic orientation, and preliminary active control, achieved through integration with magnetic tunnel junctions, MTJs.

The novelty and significance of this work are considerable. The authors introduce a fundamentally new device concept that enables chirality-selective transport control in a topological semimetal, which could have major implications for future quantum electronic devices. The experimental approach is rigorous: multiple harmonic responses (1ω , 2ω , 3ω) are measured systematically, with a clear current threshold behavior and a careful symmetry analysis of the nonlinear Hall effect. The revision expands the dataset significantly, with results from 23 devices and multiple geometries, strengthening the case for reproducibility. Additional anomalous Hall measurements convincingly disfavor trivial-state mechanisms and bolster the claim that the observed effects arise from topological fermions. Importantly, the authors now acknowledge the limitations of their semiclassical modelling, explicitly discuss possible trivial contributions, and mark their discussion of Fermi-arc mediated coherence as a hypothesis. Finally, the inclusion of MTJ-based devices demonstrates a first step toward tunability, providing proof-of-concept for an actively controllable valve.

However, some limitations remain. The demonstration of active control is promising but still preliminary, with no systematic characterization of switching reproducibility, fidelity, or scalability. While the evidence presented makes trivial contributions unlikely, they cannot be strictly ruled out, and the manuscript would benefit from an explicit acknowledgment of this residual ambiguity. The reproducibility of results is now supported by a larger dataset but remains mostly documented in the Supplementary file, while the main text still focuses on single-device examples. Although the temperature dependence of the nonlinear responses and quantum interference is now provided, a quantitative extraction of coherence length or phase-breaking rate is still lacking. Finally, while the semiclassical framework is acknowledged to be limited, the paper would be stronger if some form of effective modelling or numerical estimate were offered to connect the observed signals more

quantitatively to the underlying quantum metric and Berry curvature.

The manuscript would benefit from a few final refinements before publication. First, the authors should clearly state in the main text that while trivial-state contributions have been systematically examined and are strongly disfavored, they cannot be mathematically excluded. This would appropriately temper the strength of the claim. Second, the authors should consider including a summary table or figure in the main text quantifying reproducibility across all devices, showing, for example, the threshold current, nonlinear Hall amplitude, and interference visibility, rather than relegating these data to the Supplementary material. Third, the language describing the MTJ-based active control should be softened to emphasize that these experiments represent a proof-of-concept demonstration rather than a fully operational valve technology, and the authors could briefly discuss open questions such as switching speed and reproducibility. Finally, the discussion could briefly highlight the absence of a quantitative coherence length extraction and the need for more advanced theoretical modelling, positioning these as important directions for future work.

The revised manuscript is a significant improvement over the original manuscript. With the clarifications suggested above, it will be a strong candidate for publication in Nature. The combination of chirality separation, orbital magnetization control, and quantum interference in a single platform represents a compelling advance for topological quantum electronics.

Referee #2

(Remarks to the Author)

The revised manuscript has been greatly improved, and the authors' replies have addressed most of my earlier concerns in a constructive manner. The work is of high interest, and I find the central idea of a chiral fermionic valve both original and potentially impactful. Nevertheless, I still have several questions regarding the work's broader impact and technical validation, which need further clarification:

1. The supplementary figures (e.g., Fig. S4) show the current dependence of first-, second-, and third-order responses, but the expected quadratic ($V2\omega \propto I^2$) and cubic ($V3\omega \propto I^3$) scaling are not explicitly demonstrated. Presenting log-log plots with fitted slopes would provide a clearer validation of the power-law relations and help exclude alternative nonlinear mechanisms.
2. This paper demonstrates a novel device concept with potential applications in "topological quantum electronics". It would be highly valuable for the authors to elaborate on and discuss more concrete application scenarios. For example, could this valve structure serve as a fundamental building block for a specific type of memory or logic gate, a highly sensitive magnetic/orbital field sensor, or a component in a quantum information processing architecture?
3. The authors mention that the underlying mechanism is almost independent of relaxation time, which enables the observation of quantum interference over a length scale exceeding 15 μm . From a fundamental perspective, a critical question emerges: What is the expected scalability of this phenomenon? Is there a theoretical upper limit to the device size in which this coherence can be sustained? Discussing whether this chiral transport and interference could, in principle, exist on a macroscopic scale would provide profound insight into the robustness and ultimate boundaries of the observed quantum effects.

Referee #3

(Remarks to the Author)

I am generally satisfied with the authors' revisions and recommend the manuscript for publication after several minor revisions. Specifically, there remain a few points that need to be commented on to enhance the legibility of the manuscript.

1. Orbital angular momentum (OAM). The current discussion of OAM is vague, and its mathematical relation to quantum geometric quantities has not been clarified. Given the central role of OAM in the manuscript, I strongly suggest the authors provide a better argument.
2. Time-reversal symmetry and quantum metric. In my previous review, I raised concerns regarding the relation between time-reversal symmetry (TRS) and the quantum metric. The existence of a finite quantum metric does not depend on the breaking of TRS, while time-reversal invariant systems can still host nonzero quantum metric. What requires TRS breaking (in addition to inversion breaking) is the manifestation of intrinsic nonlinear Hall effects induced by the quantum metric. The authors' reasoning that a finite quantum metric itself is evidence of TRS breaking needs to be revised.
3. Mechanism of nonlinear Hall effect. The attribution of the observed nonlinear Hall response to the quantum metric remains insufficiently discussed. The authors are encouraged to provide at least a more comprehensive discussion on possible mechanisms of the nonlinear Hall effect coexisting with the quantum metric, such as those in arXiv:2410.04995.

Peer Review File

Manuscript Title: Chiral Fermionic Valve driven by Quantum Geometry

Author's response to initial comments

We sincerely thank the referees for their valuable time in thoroughly reviewing our work. We are pleased to receive positive comments and insightful questions. We have attempted to address these questions carefully and have incorporated the feedback we received into the manuscript. We have also improved our understanding in the process of addressing these questions and concerns. We aim to convey these improvements and address the referees' questions through our responses below.

Reply to Referee #1:

This manuscript reports a novel mechanism for chirality-based filtering of fermionic currents using the quantum geometry of multifold topological semimetals, demonstrated in PdGa. The authors design a three-arm device that spatially separates topological chiral fermions without magnetic fields, detect their orbital magnetization, and demonstrate their phase coherence via Mach-Zehnder interferometry. The work introduces the concept of a “chiral fermionic valve”, highlighting the control and coherence of chiral topological transport.

The study is bold, imaginative, and potentially transformative for topological quantum electronics. The experimental design is clever, and the results suggest new device paradigms. However, the manuscript also has areas where claims are overstated, and further clarifications or evidence are needed.

The idea of using quantum geometric curvature (Berry curvature and quantum metric tensor) to achieve chirality filtering without magnetic fields is conceptually significant. In addition, the three-arm device and the Mach-Zehnder geometry are well-conceived and effectively probe chiral separation and coherence. It is also remarkable that the measurements across different harmonics (1ω , 2ω , 3ω) and their dependence on current and field orientation are thorough and observation of interference oscillations (Fig. 5) is impressive, suggesting long-range phase coherence in chiral topological states.

However, this manuscript is not free of major concerns.

We thank the Referee for the positive and encouraging remarks and for acknowledging the novelty of our work. As the referee recognised the use of the chiral fermionic valve in topological quantum electronics, we are currently actively pursuing that goal. We will start our discussion with question 2, since we have referenced its results in the response to question 1.

The following are a few examples:

2. The term "valve" implies a degree of tunability or switchability (e.g., as in a transistor or diode). While spatial separation is shown, active control (beyond changing current magnitude or direction) is limited. This weakens the analogy to classical “valves”.

Fig. R1 | Passive control of the chiral fermionic valve. *a-c* Comparison of the third-order (a(i)-c(i)) and second-order (a(ii)-c(ii)) responses in the left and right arms of the device in the ' I^Γ On' (left column), ' I^R On' (middle column) and 'Valve Off' (right column) positions. The electrical configuration used to measure these responses was the same as shown in Fig.2.

We thank the Referee for asking this very insightful question. We will first experimentally demonstrate the passive control of the valve by controlling the relative contribution of $\Delta\Omega_{\mathbb{Q}}^L$ and $\Delta\Omega_{\mathbb{Q}}^R$ in the OAM dipole using distinct symmetries of topological Fermi-pockets. Then we will explore a strategy to achieve active control of the valve. We can define four working states of the valve, depending on the relative magnitude of chiral currents in different arms. 'Valve On' corresponds to the condition when chiral currents of similar magnitude are generated in their respective arms, ' \mathbb{Q}^Γ On' (' \mathbb{Q}^R On') is a state when a chiral current is predominantly generated in the right (left) arm of the device due to NLH contribution from Γ (R) Fermi-pocket. Finally, the 'Valve Off' state in which chiral current generation is suppressed in both arms. While the symmetry of a crystal is defined by its space group, the symmetry of a Fermi pocket is determined by the local symmetry of the band structure at a given k -point, which can differ from the overall crystal symmetry. Thus, a crystal lacking mirror planes can possess Fermi pockets at high-symmetry k -points that locally exhibit mirror symmetry.¹ The chiral currents generated in different arms of the device depend on the magnitude of $\mathbb{Q}_{\mathbb{Q}}^L$ and $\mathbb{Q}_{\mathbb{Q}}^R$, as given in Eq. 1. We can tune the relative magnitude of $\Delta\Omega_{\mathbb{Q}}^L$ and $\Delta\Omega_{\mathbb{Q}}^R$ by choosing the crystallographic direction of the applied current guided by their distinct mirror planes. The device mentioned in the manuscript is in a 'Valve On' position. **Fig. R1(a)-(c)** shows three other valve positions: ' \mathbb{Q}^R On', ' \mathbb{Q}^Γ On', and 'Valve Off' in devices made in different crystallographic directions. The details about the crystal orientation can be found in Table R5(a) in response to the Referee's question 5, where we categorized different fabricated devices into four valve positions. **Fig. R1a(i)-(ii)** show the third-order and second-order responses, respectively, in the left and right arms of the device in the ' \mathbb{Q}^R On' position. We notice that the magnitude of the nonlinear responses in the left arm is much greater than in the right arm. It is due to the presence of a mirror symmetry along the xz -plane Fermi-pocket at \mathbb{Q} , which diminishes the magnitude of $\mathbb{Q}_{\mathbb{Q}}^L$ along y . Meanwhile, the mirror symmetry along the xz -plane remains broken for the Fermi pocket at \mathbb{Q} . The opposite happens in the device in ' \mathbb{Q}^Γ On' position, as shown in **Fig. R1b**. In this case, nonlinear currents

are predominantly generated in the right arm of the device. Finally, **Fig. R1c** shows the nonlinear responses in a device in ‘Valve Off’ state, where the generation of chiral current is suppressed in both arms. We call this strategy to control the valve position ‘passive’ because the valve turnability is linked with the intrinsic quantum geometry of the topological bands rather than the experimentally controlled parameter.

We further aimed to achieve active turnability of the valve, which would significantly enhance its practical use. As discussed in the main text, the chiral fermionic current exists due to the preferential occupation of R (\mathbb{Z}) Fermi pocket in the left (right) arm of the device. Thus, the magnitude of chiral currents can be controlled by tuning the occupational imbalance between two Fermi pockets. We need a mechanism to modulate the relative occupancy to achieve active control. Typically, we rely on electromagnetic fields to modulate the occupancy of Fermi pockets. We cannot use local electrostatic gating since these chiral currents exist in the bulk metal. Meanwhile, applying an external magnetic field would not allow tuning the chiral currents locally in both arms. We fabricated a magnetic tunnel junction (MTJ) with in-plane magnetization on top of individual arms to locally probe the influence of the magnetic field on chiral currents. MTJ locally breaks the time reversal symmetry without directly influencing the current transport in the bulk. The MTJ deposition requires a smooth surface without sudden height change for the thin insulating MgO layer to create an efficient tunnelling barrier. The fabrication of MTJ on top of the 2-3 μm lamella is nontrivial because it would create a sudden height step between the substrate and the lamella. We developed a new fabrication technique to overcome this challenge. We first milled a trench in the substrate with the same geometry and thickness as the lamella. We then meticulously placed the lamella inside the trench so that the surface of the lamella was at a similar height as to the substrate. Then the MTJ stack was sputtered on top of the lamella, and the MTJ electrodes were patterned using low-energy ion-beam etching in FIB. **Fig. R2(a)** shows the false-coloured SEM image of the prepared device with MTJ electrodes deposited on top of the left and right arms. The lamella was prepared in the ‘Valve On’ position.

Fig. R2 | Active control of the chiral fermionic valve. (a) (top) False-coloured SEM image of the device prepared by placing the lamella inside the trench made in the substrate. (middle) Schematic of the MTJ stack deposited on top of the device. (bottom) Top view of the device. The scale bar is 10 μm . **b(i)-(ii)**

Dependence of the third-order response in the left and right arms of the device in the MTJ configuration with magnetization direction along the X and Z directions at a 77.77 Hz frequency of the applied current. c(i)-(ii) The variation of the third-order response in left and right arm with the frequency of the applied current. The magnetization of the MTJ was along X. The electronic configuration was the same as shown in Fig.4 of the manuscript.

In the first set of experiments, we studied the effects of the magnetization direction of the MTJ on the chiral current of both arms. The electrical configuration to measure these responses was the same as shown in Fig. 4 of the manuscript. We will discuss two magnetization directions of the MTJ, M_x and M_z (refer to the coordinates shown in the inset of the image). M_x and M_z represent directions when the MTJ's magnetization is perpendicular and parallel to the current-induced orbital magnetization. **Fig. R2b(i)-(ii)** show the $\mathbb{Z}_{3\mathbb{D}}$ responses at 77.77 Hz in the left and right arms, respectively, with M_x and M_z magnetization. We can observe from **Fig. R2b** that the chiral current has the same order of magnitude in the left and right arms at M_x orientation. This represents the valve in the 'On' position. We observed that the nonlinear response of the right arm switches sign when the MTJ magnetization was switched from M_x to M_z ; meanwhile, the $\mathbb{Z}_{3\mathbb{D}}$ response in the left arm remains similar. Notably, these measurements were performed in the absence of an external magnetic field. The magnetic field was only used to switch the magnetization direction of the MTJ. **Fig. R2b** shows that we can actively tune the valve from the 'Valve On' position to the ' \mathbb{R} On' position by switching the MTJ magnetization from M_x to M_z . In the second experiment, we varied the frequency of the applied current to study the interaction of the chiral current with the MTJs' magnetization by measuring the inductance impedance. **Fig. R2c** shows the dependence of $\mathbb{Z}_{3\mathbb{D}}$ responses at different frequencies of the applied current with MTJ in M_x configuration. We notice from **Fig. R2b(ii)-c(ii)** that the impedance response of the $\mathbb{Z}^{\mathbb{R}}$ is not significantly changed from 77.77 Hz to 47 Hz, while the impedance response of the $\mathbb{Z}^{\mathbb{D}}$ is reduced by half. From **Fig. 2c(i)**, we observe that the magnitude of $\mathbb{Z}^{\mathbb{D}}$ further reduces by an order of magnitude upon going to the frequency of 23 Hz. Meanwhile, the $\mathbb{Z}^{\mathbb{R}}$ response, even though diminished, has the same sign and the order of magnitude. Through **Fig. R2c**, we show that the valve can be tuned from 'Valve On' position to ' $\mathbb{Z}^{\mathbb{R}}$ On' position by changing the frequency of the applied current from 77.77 Hz to 23 Hz with the MTJ in M_x configuration. In these two experiments, we have shown a strategy to pursue active turnability of the chiral fermionic valve by its integration with an MTJ. The explanation behind these results needs a deeper understanding of the interaction between the chiral current and magnetization dynamics, which is currently not the primary focus of our study. However, we have discussed a possible mechanism as our hypothesis in response to the Referee's question on the role of Fermi arcs in chirality coherence.

1. The authors attribute current directionality to topological states based on anomalous velocity, but it's not clearly disentangled from possible trivial contributions. Evidence ruling out trivial state effects is circumstantial.

We thank the referee for raising this important question. We meticulously looked for trivial-states-based intrinsic and extrinsic mechanisms that may give current directionality similar to topological states-based anomalous velocity. In our study, we show two distinct nonlinear Hall (NLH) responses in the outer arms of the device. Therefore, the probable trivial states-based mechanism should produce a dipolar field to generate distinct NLH currents in both arms. These distinct NLH currents are also locked to a specific arm, since they choose the same arm when current direction is reversed in an AC signal. Therefore, the transverse velocity should be even in the applied electric field. In pursuit of these criteria, we will discuss current directionality based on spin-galvanic effect stimulated by our unique device

geometry, driven by the Rashba-split trivial bands under nonequilibrium. However, we will also then provide experimental evidence to rule it out. We also want to mention that we intentionally designed our device to avoid contact or shape anisotropy-based effects.² Also, emphasis on reducing the contact resistance (< 2 Ohm with 30 nm of Au) was given to avoid joule heating effects at the currents measured in our study.³

Fig. R3 / Anomalous Hall effect. **a** The Rashba split trivial bands in the presence of spin-orbit coupling. **b** The spin current galvanised due to the presence of $Q_{\alpha\gamma}$ of the spin-split trivial band. **c** The electronic configuration used to measure the Hall responses. **d** The Hall response of the device in the ‘Valve On’ state. **e-f** The relative variation for the Hall response with the magnetic field orientation in the device with ‘ I^R On’ and ‘ I^I On’ positions. These responses were measured using 400 μ A of applied current at 2 K.

PdGa belongs to a gyrotropic class with the tetrahedral chiral point group 23 (T), which allows for a Dresselhaus-type spin-orbit coupling.⁴ The spin-orbit coupling can cause splitting of the trivial band (and topological bands). The applied electric field creates nonequilibrium spin polarization of both spins, as shown in **Fig. R3(a)**. The current applied along z would create a spin polarization for both spins along z . In 2002, Ganichev et. al. proposed the spin galvanic effect, in which electric current was produced due to spatially uniform nonequilibrium spin polarization.⁵ Empirically, the electric current density (\mathbb{J}) is linked with the electron’s average spin (\mathbb{S}) by $\mathbb{J} = \sum_{\alpha\beta\gamma} \mathbb{Q}_{\alpha\beta\gamma} \mathbb{S}_{\alpha\beta}$, where \mathbb{Q} is the second-rank pseudotensor of a gyrotropic crystal. These currents are semiclassically modelled using spin-dependent scattering asymmetry. In the conventional spin galvanic effect, the current is galvanized due to the asymmetry in spin-flip scattering, which requires a spin population imbalance of a particular spin. However, opposite spin currents can also be galvanized due to the asymmetry in skew scattering, as in the case of inverse spin Hall effect, as schematically shown in **Fig. R3(b)**. Similar to the spin galvanic effect, the inverse spin Hall effect also generates a dipolar term proportional to $(\mathbb{S} \cdot \mathbb{S})$ due to spin-dependent correction of the Fermi-Dirac distribution ($\mathbb{F}_{\mathbb{S},\mathbb{S}}$).⁶ For non-zero $\mathbb{Q}_{\alpha\beta\gamma}$, this would generate currents of opposite spins in different arms of the device, instead of the spin Hall voltage measured in a conventional Hall device. The spin currents generated due to inverse spin Hall effect (or spin galvanic effect) can give the desired current directionally solely due to the trivial bands of PdGa. Such a filtration of spin current into different arms is conceptually similar to the filtration of the chiral fermions due to electric field-induced quantum geometry.

The magnitude of opposite spin currents galvanized into the outer arms would depend on the size of the Fermi surface and $\delta f_{k,S}$. These parameters are similar for both spin currents, since they originate from the same trivial band's Fermi-surface. Therefore, one would expect that the *relative* magnitude of current in both arms would remain similar. It would be irrespective of the crystallographic direction of applied current, even when the magnitude of individual spin current may vary due to anisotropic skew scattering. In our discussion with Fig. R2, we have shown that we can tune the relative magnitude of the generated current in both arms by passing current in different crystal directions. Therefore, the spin currents galvanised due to $\mathbb{E}_{\mathbb{P}\mathbb{P}}$ of trivial bands cannot explain the differences in the relative magnitude of currents measured in different devices. We will now provide the second evidence by ruling out the involvement of the $\mathbb{E}_{\mathbb{P}\mathbb{P}}$ -like tensor from the trivial bands. We measured anomalous Hall response in three devices with crystallographic orientation corresponding to ‘Valve On’, ‘ $\mathbb{P}^{\mathbb{R}}$ On’ and ‘ $\mathbb{P}^{\mathbb{F}}$ On’ positions. **Fig. R3(c)** shows the electrical configuration used to measure the Hall responses. The magnetic field was rotated in the yz plane, where $\mathbb{P} = 90^\circ$ corresponds to the field along the direction of applied current z . **Fig. R3(c)** shows the relative change in the magnitude of the Hall response on magnetic orientation in a device with ‘Valve On’ position with respect to $\mathbb{P} = 90^\circ$. The response in this valve position resembles the response expected from a trivial Hall effect. $\Delta\Omega_{\mathbb{P}}^{\mathbb{F}}$ and $\Delta\Omega_{\mathbb{P}}^{\mathbb{R}}$ have a similar magnitude contribution to the OAM dipole in the ‘Valve On’ position. Therefore, there is no anomalous Hall response due to the absence of any net magnetization. The Hall response was also similar in the device in ‘Valve Off’ state, which is expected due to the absence of the OAM dipole itself along the y -axis. **Fig. R3(e)-(f)** show the relative change in the magnitude of the Hall response with the magnetic field orientation in the device in ‘ $\mathbb{P}^{\mathbb{R}}$ On’ and ‘ $\mathbb{P}^{\mathbb{F}}$ On’ position. respectively. These responses show the presence of an anomalous Hall response with distinct symmetries. In ‘ $\mathbb{P}^{\mathbb{R}}$ On’ position, a net magnetization is present because $\Delta\Omega_{\mathbb{P}}^{\mathbb{R}}$ has larger contribution in the OAM dipole than $\Delta\Omega_{\mathbb{P}}^{\mathbb{F}}$. The symmetry of the Hall responses matches the symmetry of the longitudinal response of chiral current discussed in Fig. 4b(i) in the main manuscript. Meanwhile, $\Delta\Omega_{\mathbb{P}}^{\mathbb{F}}$ contributes more to the OAM dipole in ‘ $\mathbb{P}^{\mathbb{F}}$ On’ position; therefore, the anomalous Hall response captures its two-fold symmetry. The three distinct anomalous Hall responses observed in our study cannot be explained by $\mathbb{E}_{\mathbb{P}\mathbb{P}}$ -driven currents. Our results show that the observed two distinct Hall responses can only be explained by considering the $\Delta\Omega_{\mathbb{P}}^{\mathbb{F}}$ and $\Delta\Omega_{\mathbb{P}}^{\mathbb{R}}$ contributions coming from two different topological Fermi-pockets.

3. Several theoretical derivations (e.g., Eq. 1 and subsequent semiclassical analysis) are presented without discussing limitations of approximations, especially in systems with spin-1 and spin-3/2 quasiparticles.

We thank the referee for raising the important aspect regarding the limitations of the semiclassical equation. The semiclassical Boltzmann approach considers electrons as adiabatic Bloch wavepackets moving in a *static* bandstructure. This assumption leads to the following limitations in modelling coherent chiral currents generated by the electric-field-induced quantum geometry –

- 1) In our study, we have shown the phase coherent transport of chiral fermions carrying orbital angular momentum. Therefore, the assumption of a localized and non-interacting Bloch wavepacket used in the Boltzmann formalism is invalid for the multifold fermions with spin - 1 and spin -3/2.
- 2) With Fig. 4, we semiclassically describe the electron carrying an orbital magnetization due to its local Berry curvature. However, we have also shown a long-range coherence of the chiral current. Therefore, the modelling of phase-coherent transport is limited by the assumption of the localized wavepackets used in the semiclassical analysis.

- 3) A standard Boltzmann formalism often assumes a single distribution function per band, effectively treating all carriers as coming from a single Fermi surface. However, we have shown chirality-selective transport due to the imbalance in the occupation of two different Fermi-pockets with opposite Chern-number. These currents of different chirality may have different scattering and relaxation dynamics, which should be incorporated in the transport equations.
- 4) One of the most important limitations regarding explaining our experimental results is the assumption of a static band structure. We have shown that the nonlinear transport responses appear only after a certain current threshold due to non-zero quantum geometry. Therefore, the electric field-induced transitions cannot be captured using the metric of the band structure at equilibrium.

4. The discussion on Fermi arcs as a mechanism for chirality coherence and protection is speculative (pp. 20-21). This section should be better supported or more clearly marked as a hypothesis.

We agree with the reviewer that a systematic study needs to be performed to support the claim of the Fermi-arcs' role in chirality coherence. Therefore, we added text to mark it as a hypothesis. However, we would like to briefly discuss two experiments conducted towards proving this hypothesis. In these two experiments, we locally probed the influence of time-reversal and inversion symmetry on the surface on how they affect the transport of chiral current in the bulk. The experiments are as follows –

- 1) Our initial attempt to achieve active turnability, discussed earlier, was using an external magnetic field. However, the magnitude of the nonlinear response of the chiral current didn't vary with the static magnetic field (refer to **Fig. R14**). The observed behaviour is likely because the quantum metric induced nonlinear conductivity does not scale with the scattering time that sets ordinary Drude conductivity.⁷ As a result, we could observe a clear quantum interference signal till 6 T (refer to **Fig. R10**). Hence, we aimed to achieve active turnability through Fermi-arcs in the MTJ incorporated device discussed earlier. The Fermi-arcs in topological matter provide a direct pathway connecting the two topological bands with opposite Chern numbers. The control over the fermionic transport over these pathways would enable tuning the occupation imbalance between the two Fermi-pockets. Our initial results, as discussed in **Fig. R2(b)**, suggest promising prospects in that direction. We have shown in **Fig. R2b(ii)** that we can switch the chiral current character by changing a surface property. This indicates that the surface property can influence the chirality transport in the bulk.
- 2) In a conventional Hall bar device (without the left and right arms), we shorted the side walls of the conducting channel through the Au bridge. The nonlinear longitudinal responses were not observed in this configuration. Then, we etched the Au bridge atop the channel to disconnect the side walls. Upon measuring the same device without the Au bridge, we observed the appearance of the longitudinal non-linear responses. It showed that even though the Au bridge transversely shorted the side walls, the longitudinal nonlinear transport of the complete channel was affected. The lamella was prepared in 'On' position for this device.

In these experiments, the surface-dependent bulk transport led us to hypothesize the involvement of Fermi-arcs in chirality transport.

5. Only a few devices are presented. While it is stated that multiple devices were made, results shown focus on single examples. Statistical robustness is missing.

a

Device name	Z-direction	X-direction	$\phi = \tan^{-1} \frac{V_{3\omega}^R}{V_{3\omega}^I}$
QG20	[0 $\bar{1}1$]	[100]	38.65°
QG13	[100]	5° away from [0 $\bar{1}1$]	49.43°
QG09	[100]	8° away from [011]	4.84°
QG18	[011]	[100]	OFF
QG11	[100]	8° away from [010]	83.88°

b
Fig. R4 | a Table showing the crystal orientation of the devices fabricated in different valve positions. The current is applied along z , and the NLH currents are collected along x . **b.** Schematic showing the devices mentioned in the table categorized into different valve positions. The valve positions are quantified using the parameter ϕ . The third-order voltages were measured at 80 μA to calculate ϕ . The crystal is well oriented along [100], therefore the error margin while fabricating devices is within 1-2 degrees. However, the crystal is not well aligned along other directions; thus, the margin for error is a bit higher, around 4-5 degrees.

We have made 23 devices in different geometries and crystal orientations. The five focal crystal orientations are presented in **Table R4a**. We have quantified the position of the valve using the term $\phi = \tan^{-1} \frac{V_{3\omega}^R}{V_{3\omega}^I}$. $\phi = 45^\circ$ represents ‘Valve on’ position, when there are equal magnitudes of the NLH currents from the Γ and R Fermi-pockets into the right and left arm, respectively. $\phi = 0^\circ(90^\circ)$ represents ‘ I^Γ On’ (‘ I^R On’) position, where NLH currents mainly arise from the Γ (R) Fermi-pocket into the right (left) arm. We made four devices in a three-arm geometry near to the ‘Valve on’ position. All four devices show the same experimental features discussed in the manuscript, namely, the appearance of distinct NLH responses of similar magnitude in different arms after the current threshold and their distinct symmetries of first-order responses with magnetic field orientations, as discussed in Fig.3 b(i)-c(i). The absolute value of the nonlinear response varied with the device’s thickness. The modulations of the third-order responses discussed with Fig.3 b(ii)-c(ii) were observed in two of these devices. A good signal-to-noise ratio is needed to measure the nonlinear responses in the presence of the magnetic field, which we posit is a limiting factor in our PPMS system. We made three Mach-Zehnder interferometer devices with a ‘Valve on’ crystallographic position. All three showed oscillations in third-order response with applied current and magnetic field, as discussed in Fig. 5. We have made a few other MZI devices in different geometries; kindly refer to the response to question 9 of Referee 2 for more details.

Fig. R5 | (a) Schematic showing the crystal axis of devices presented in (b), (c), and (d) made by rotating the crystal axis \square away from [011] towards the ‘Valve Off’ position. (b) Relative variation of first-order response in the ‘Valve On’ position. The data presented is the same as discussed with Fig. 4b(i)-c(i). c-d Relative variation of first-order response in device with α values of 25° and 30° respectively. All these data are measured at 2 K in a PPMS system.

We showed with passive control that we can controllably turn off the NLH responses discussed in Fig. 2-3 of the manuscript. **Fig. R5** shows the relative variation of the first-order response with the magnetic field orientation in different devices fabricated by rotating the crystal axis away from the ‘Valve On’ state. We will show that the distinct responses of the chiral current discussed in Fig. 4, become two-fold symmetric in both arms in the ‘Valve Off’ state. **Fig. R5(a)** shows the schematic of the \square -degrees rotated crystal orientation relative to the crystal axis of the device discussed in the manuscript. **Fig. R5b** shows the same data discussed in the manuscript presented again for comparison. **Fig. R5(c)-(d)** show data from the devices with \square values of 25° and 30° . The response in the left arm varies considerably upon rotating the crystal axis along the 100 axes, while the response in the right arm remains similar. The crest near 30° and 300° in the response in the left arms turns into peaks, while the peak at 180° develops a crest. Upon further rotating the crystal axis towards the ‘Valve Off’ position, the peaks 30° and 300° move towards 90° and 270° respectively and the crest dip at 180° dips further. At the ‘Valve Off’ position, the responses in both of the arm are two-fold symmetric. (refer to Fig. R6(d)-(e) for reference)

6. The effect of temperature on coherence length, scattering, and nonlinear responses is not explored in depth. This is critical to understand the practical viability of such devices.

Fig. R6 | Temperature dependence of the NLH responses and quantum interference. *a-c* Variation of the first, third, and second-order responses in one of the devices' arms. *d-e* Relative variation of the first-order response in the right and left arm of the device, respectively, with magnetic field orientation at different temperatures with an applied current of 400 μA . The magnetic field of 2 T was rotated in the xy plane, as discussed in Fig. 4 of the manuscript. *f* Temperature dependence of the FFT amplitude of the magnetic fields induced oscillation of the third-order response. Note that these data, with the exception of the data in (f) were measured in a PPMS system (base temperature 2 K).

We first discuss the effect of temperature on the non-linear responses. **Fig. R6(a)-(c)** show the log-scaled temperature dependence of the first-order, $\mathbb{V}_{2\omega}$, and $\mathbb{V}_{3\omega}$ longitudinal responses in one of the arms of the device. **Fig. R6(a)** shows that the first-order response decreases with temperature till 15 K after which it starts to saturate. Upon cooling further, the $\mathbb{V}_{3\omega}$ and $\mathbb{V}_{2\omega}$ responses start to appear, as shown in **Fig. R6(b)-(c)**. The appearance of the $\mathbb{V}_{3\omega}$ response is concurrent with the observed upturn in the first-order response, as expected from Eq. 1 of the main manuscript. The $\mathbb{V}_{3\omega}$ and $\mathbb{V}_{2\omega}$ responses start to appear below 3.4 K and 14 K, respectively. The exact transition temperature varied between devices, with different device geometries and crystallographic orientations. However, the concurrent appearance in the upturn of the first and third-order responses was always observed in all the measured devices, not in the 'Valve Off' position. Data at lower temperatures were measured in a Bluefors system which showed that the nonlinear responses start to saturate below 1 K. Note that these data are not shown because they had a considerably better signal-to-noise ratio due to the low-noise filters present in the Bluefors system. **Fig. R6(d)-(e)** show the relative modulation of the first-order responses with magnetic field orientation in the right and left arm, respectively, at different temperatures. The modulation in the different arms remains of opposite signs below 100 K even in the device with 'Valve Off' position. After 100 K, the change in the sign of the modulation in the right arm was observed. Thereafter, both arms share the same sign of modulation, albeit of different magnitudes. Above 200 K, the modulation with magnetic field orientation starts to disappear in both of these arms.

Fig. R6f shows the temperature dependence of the FFT amplitude of the $\mathbb{V}_{3\omega}$ oscillation with the magnetic field. The amplitude of the oscillation decays slowly with temperature below 1 K, after which it starts to decrease more sharply with temperature. The amplitude decay trend matches the temperature dependence of the magnitude of the $\mathbb{V}_{2\omega}$ shown in **Fig. R6c**. We strongly believe that the

FFT amplitude is linked with the quantum metric magnitude. The $\mathbb{Z}_{2\mathbb{Z}}$ response is the direct indicator of quantum metric response. Assuming amplitude decay solely due to an inelastic mechanism would not be accurate since the quantum metric doesn't scale with the scattering time in Drude conductivity. Therefore, the amplitude decay was not correlated to the phase coherent length because the mechanism of decoherence of chiral fermions is not completely known. We also tried to measure the amplitude of the oscillation of the $\mathbb{Z}_{3\mathbb{Z}}$ response with applied current shown in Fig. 5(b). We observed that the oscillation period changed with temperature, and the FFT amplitude broadened into multiple peaks. Hence, tracking the FFT amplitude with temperature was not trivial. It may be due to variation of current-induced magnetization with temperature, which influences the oscillation period.

A few minor concerns:

1. Several sentences are verbose and could benefit from tightening. For example, the abstract can be made crisper for clarity.

We have now tightened the abstract for clarity. We have also made a few changes in the main text.

2. Repeated use of phrases like “remarkably” or “demonstrate” makes some claims feel overstated.

We thank the referee for the suggestion, which we have now incorporated into the main text.

3. Several claims about chiral coherence, magnetization, and device implications would benefit from additional references, particularly to related work in non-linear Hall effects and topological electronics.

We thank the referee for the suggestion. A few of the main references added are as follows -

1. Kawamura et al. showed the coherent chiral charge pumping between the edges of a Chern insulator through the bulk of the device in a Corbino disk geometry.⁸
2. A theoretical study on observing fractional statistics of a Laughlin quasiparticle in a three-terminal Aharonov-Bohm ring by measuring the telegraph noise.⁹
3. A study by Taktak et al. shows the large quantum coherence of anyons in an electronic Fabry-Pérot interferometer in the fractional quantum Hall effect regime.¹⁰
4. We have also added some references discussing intrinsic contribution to non-linear Hall effect from Bloch-state orbital magnetic moments and some work discussing the device proposition of topological materials towards topological electronics.¹¹⁻¹³

All in all, the manuscript has the potential to be a high-impact publication in Nature, but it currently overstates its conclusions and needs stronger evidence or clarification in key areas. The novelty of the chiral fermionic valve concept is substantial, but its practical tunability and control are still preliminary. A deeper discussion of limitations, more evidence excluding trivial contributions, and clearer exposition of theoretical assumptions will significantly improve the manuscript.

We sincerely appreciate the questions asked, which have challenged us to go further than our proposed device concept. We have made the following changes to the manuscript to address the Referee's concerns -

1. We have strengthened our claim of the chiral fermionic valve by adding a section on passive turnability in the manuscript. We attempted to achieve active turnability through Fermi-arcs by incorporating MTJ into our device geometry. Even though we have shown its active turnability, we believe it requires a separate examination to study further the rich interaction between the

chiral current and magnetization dynamics. However, we are willing to incorporate it in our study based on the Referee's recommendation.

2. We have discussed the limitations of the semiclassical analysis in modelling chiral currents due to electric-field-induced quantum geometry in the manuscript.
3. We have added a discussion in the Supplementary file on possible trivial states based on current directionality in the context of spin galvanic effect, with experimental analysis to rule it out.
4. We have added a section on discussing the temperature effect of NLH responses and quantum interference in the Supplementary file.

Reply to Referee #2:

In this manuscript, Dixit et al. fabricated the nanoscale Hall device from a single crystal of PdGa to separate the currents with opposite fermionic chiralities by exploiting the quantum geometry of topological bands. They then observed the quantum interference of these chiral currents with mesoscopic phase coherence. The demonstration of mesoscopic coherent transport of chiral fermions is crucial to realizing functional quantum electronics. Their strategy to separate the left- and right-handed chiral fermionic currents and to interfere with them is very attractive. These chiral fermionic devices pave the way for practical applications that harness the chiral degree of freedom in topological quantum materials. While their experimental results are very interesting, there are several aspects that need further clarification.

We thank the referee for acknowledging the main findings of the paper and for the positive comments about our work and its practical applications.

(1) As reported in this work, the nonlinear Hall (NLH) effect exhibits an obvious threshold current of around 55 μA . Above this threshold current, both the second and third NLH signals appear. They attribute this transition to the appearance of the NLH currents. However, it is not clear to me why the NLH currents can induce such a transition. Moreover, there is a jump in AC resistance at a similar current. The authors claim that this resistance jump was not observed in DC measurements. What is the resistance difference between AC and DC measurements?

Fig. R7 | **a** SEM image of the device made by removing the left arm from the three-arm geometry discussed in the manuscript. **b**. Variation of the first-order response with applied current in the right arm of the device. The electrical configuration used to measure the response is given in the inset. **c**. Dependence of the first-order transition on the out-of-plane magnetic fields in a three-armed device. The crystal direction of the fabricated device is mentioned in Table R4(a) with device name QG11.

We thank the Referee for raising the important question regarding the origin of the transition. We show in the manuscript that the NLH currents appear after the current transition, when the quantum

metric \mathcal{D} becomes non-zero. The non-zero \mathcal{D} is typically observed in PT -symmetric systems with simultaneously broken P and T symmetry. We have shown in our study that these conditions can also be fulfilled by an electric-field-induced OAM dipole. It acts like an orbital analogue of spins in an antiferromagnetic system where non-zero \mathbf{g} was previously observed.⁷ If we understood correctly, the Referee's question is regarding the origin of such a transition, when \mathcal{D} becomes non-zero. We probed this question to understand whether the transition is an intrinsic property of the bandstructure under nonequilibrium or is it device geometry dependent. We will discuss two sets of experiments to answer this question. **Fig. R7(a)** shows the device geometry without the left arm, while keeping the middle and right arms. The other device properties, including the crystallographic directions of the applied current, device width, and thickness, were similar to those discussed in the main text for the three-arm device. **Fig. R7(b)** shows the first-order response of the current flowing in the right arm of the device. The first-order response shows a similar jump at around 55 μA . Next, we fabricated the device in a three-arm geometry with a different crystallographic orientation, while keeping the other device geometry parameters similar. **Fig. R7(c)** shows the first-order response for the current going into the left of the device at different magnetic fields applied along x (refer to Fig. 1 for the coordinate axis). In this device, the transition occurred at around 44 μA . The transition occurs at the same current even under the magnetic field of 20 mT, however, the jump in the response is diminished. The appearance of the transition at the same current even in the presence of magnetic field-induced scattering indicates that the transition is electric-field-induced rather than current-induced. The transition fades away at 40 mT. These experiments indicate that the electric-field induced transition is an intrinsic property of the band structure at nonequilibrium. The DC resistance measured in different devices was similar to that of AC in the linear regime. The AC impedance was higher after the current threshold when the nonlinear responses started to appear due to the inductive impedance of the chiral fermionic current.

(2) What is the relationship between the magnitude of NLH signals and applied currents?

Conventionally, the relationship between $V_{3\omega}$ ($V_{2\omega}$) and I_{ω} follows the cubic (square) law.

The NLH responses start to appear after a certain current threshold; therefore, our tried polynomial fits were inaccurate. However, after the transition the third and second order responses follow cubic and square current dependence, as mentioned by the Referee.

(3) The measured current is the total current that flows through all left, right and middle arms. Does the ratio of the currents flowing in the three arms remain the same at different currents?

In order to measure the ratio of currents going into different arms, we grounded the left, right, and middle arms of the device while measuring the current passing through. We observed an equal magnitude of currents going into the ground through these arms for all current values. However, this strategy doesn't emulate the flow of NLH currents in our three-armed geometry device. The electric-field-induced quantum geometry responsible for the NLH current is proportional to $\mathcal{D}_{\parallel}^2 \mathcal{D}_{\perp}$ (from Eq. 1 of the main manuscript). When we ground the left and right arms of the device, there is an additional Ohmic contribution to the current due to \mathcal{D}_{\parallel} , which is more dominant than the contribution from the NLH current. Therefore, the same ratio of current measured in both arms at different currents shows that the three current pathways share the same device geometry. The more suited device geometry to measure the ratio of NLH currents and the Ohmic current the middle would be the two-arm geometry discussed in **Fig. R7a**. In this case, the ratio of first-order response in the right and middle arms was between 3 and 4 in different devices of the same crystal orientation. This ratio varied with different crystallographic orientations, as expected from the anisotropic nature of the electric-field-induced

quantum geometry. Notably, such a two-arm device still doesn't exactly emulate the three-arm geometry mentioned in the manuscript, since the two-arm geometry discussed in **Fig. R7a** is not geometrically symmetric with the z -axis (direction of applied current). Therefore, there can be some trivial contributions of current into the right arm.

(4) It is noticed that in Fig. 4b(ii), the magnetic field and device temperature are significantly different from those in other panels. And there are significant differences in signal magnitude. Is there an explanation for its cause?

First, we would like to discuss the reason for the difference in the relative magnitude between the left and right arms of the device. As discussed in the main text, the modulation with magnetic field orientation is proportional to the orbital magnetization $m^{\Gamma,R}$ carried by the chiral currents in the respective arms. $m^{\Gamma,R}$ can arise due to the contribution from the Berry curvature and quantum metric. The orbital magnetization due to the Berry curvature is inversely proportional to the distance between the Fermi-level and the topological band-crossing. The band crossing at Γ sits much closer to the Fermi-level than at R in PdGa. Therefore, the magnitude of orbital magnetization due to the Fermi-pocket at Γ would be greater than R . It explains the more pronounced modulation of the chiral current in the right arm as it is preferentially carried by the topological bands at Γ .

Fig. R8 | The relative modulation of the third-order response in the left arm of the device measured at 2 K at $400 \mu A$.

Now we will address the Referee's specific question regarding Fig. 4b(ii), which shows the relative modulation in the $\mathbb{E}_{3\omega}$ response with magnetic field orientation in the left arm. The chiral current in the left arm is preferentially carried by the topological bands at R . As discussed above, the orbital magnetization due to Berry curvature of Fermi-pocket at R is much smaller than at Γ . Additionally, the nonlinear responses only start to appear below a certain temperature threshold (discussed in more detail in response to question 6 of referee 1). The transition temperature measured in the PPMS system is in the range of 2-3.5 K. The $\mathbb{E}_{3\omega}$ response starts to decay sharply with the magnetic field near to the transition temperature. Therefore, we couldn't resolve the $\mathbb{E}_{3\omega}$ response in the left arm at 2 T. We performed the measurement in the Bluefors system with a much better signal-to-noise ratio, which helped resolve the third-order response. We also cooled down the sample further below the transition temperature, after which we observed the three-fold modulation in the $\mathbb{E}_{3\omega}$ response with magnetic field orientation. However, we could measure a similar response at 2 K in a different device with a different crystallographic orientation, as shown in **Fig. R8**. We were able to measure this data at 2 K because

the relative contribution of the chiral current in the left arm is bigger in this particular crystallographic orientation. (discussed in more detail in response to question 2 of referee 1).

(5) In the interference measurement, they removed the middle arm. Is it just to mimic the Mach-Zehnder interferometer? What is the effect of the middle arm on the interference? Without the middle arm, the trivial current will flow into the left and right arms, which still induces a background electrical signal.

Fig. R9 | Oscillation in the third-order response with applied current at 3.4 K in the three-arm geometry at zero external magnetic field.

As the Referee correctly pointed out, the middle arm was removed to mimic the Mach-Zehnder interferometer (MZI). The difference in the phase acquired by the fermions in two arms of the MZI gives an interference pattern with a well-defined period. However, in the three-arm geometry, we would also need to consider the different combinations of trajectories a fermion takes in the presence of the middle arm. Therefore, the period of the oscillation would not be well defined, and the amplitude of the oscillations would be diminished. **Fig. R9** shows the oscillation in the $\mathbb{R}_{3\omega}$ response with applied current in a device with a three-arms geometry. We can observe that the oscillations are weaker compared to the interference observed in the MZI geometry, and they don't have a well-defined period. The oscillations are also on top of the background signal, which has a similar shape as the $\mathbb{R}_{3\omega}$ response shown in Fig. 2 of the manuscript. It suggests that the responses coming from different arms don't exactly cancel out, hence the oscillations are not occurring around zero after the current threshold. We observed a similar offset in the interference response even in the MZI geometry when the chiral currents in the left and right arms are not of similar magnitude. The interference data shown in Fig. 5(b) shows oscillations with the negligible offset even after the threshold current. It shows that a similar magnitude of chiral current exists in both arms. One of the questions that arises from **Fig. R9** is why the oscillations occur even below the threshold current? It will be discussed in detail with response to question 7.

(6) They observed the quantum interference of chiral current at the current range of +/-100 uA and field range of 0-1 T. Can larger currents/magnetic fields (negative field?) maintain coherence and interference? I also notice that the device temperature needs to be very low (30 mK) to observe the interference effect. Is this interference effect limited by the coherence length, which dramatically decreases with the temperature? If so, how is the relationship between the coherence length and the device temperature?

Fig. R10 | **a.** Oscillations in $V_{3\omega}$ response with applied magnetic field with a current of $30 \mu\text{A}$ at 100 mK . **b.** Expanded view of the data from **a.** **c.** The phase dependence of the oscillation with the direction of the magnetic field sweep. The magnitude of the field-sweep rate was 0.83 mTs^{-1} .

Fig. R10a shows the $V_{3\omega}$ response across the MZI measured with an applied magnetic field. The oscillations in the $V_{3\omega}$ response due to quantum interference could be clearly seen till 6 T , as shown in **Fig. 10b**. The oscillations persist in the negative magnetic fields as well. **Fig. 10c** shows the interference measured from -0.3 T to 0 T . We observed that oscillations' phase was shifted by π when the magnetic field sweep direction was reversed. It further proves the inductive nature of the chiral fermionic current. In the manuscript, we presented the data only till 1 T , because the FFT amplitude peak splits at higher magnetic fields. This is likely due to the interactions of the magnetic field with different spins of the chiral fermions. We measured the oscillations in $V_{3\omega}$ with applied current up to $150 \mu\text{A}$. We didn't go above this current value because the device temperature increases very sharply with applied current. Therefore, even at $150 \mu\text{A}$, the device temperature is $3\text{-}4$ times the base temperature. At higher device temperatures, the oscillation period doesn't remain uniform, possibly due to the temperature dependence of the current-induced magnetization. We kindly refer the reviewer to our response to question 6 of reviewer 1, where we discussed the temperature dependence of the nonlinear responses and quantum interference data. The oscillation amplitude magnitude doesn't suddenly drop with the temperature as we go from base temperature to 1 K . The amplitude starts to decay after 1 K as shown in **Fig. R6f**. The amplitude delay is related to the magnitude of the quantum metric which drives these nonlinear responses. The reason for choosing the lower device temperature and magnetic fields was that we observed oscillations with a well-defined period in this regime. At higher magnetic field and current values, additional effects influenced the oscillation period. We have updated **Fig. 5** in the manuscript to show interference data up to 2 T and dependence of field-sweep direction on the oscillation phase.

(7) This question is related to comment (1). Unlike the device with a middle arm (**Fig. 2**), there is no transition of NLH signals with respect to current in the interference device (**Fig. 5b**). The device can work at a relatively low current of $20 \mu\text{A}$. What is the difference between the two devices? Temperature or device geometry?

Fig. R11 | **a-b** Schematic showing the generation of chiral currents from different Fermi-pockets below and above the threshold current.

We thank the referee for asking about this important question. First, we would like to point out that the transition in the first-order response was still observed in the MZI geometry. However, we will discuss the important reason behind the interference of nonlinear chiral currents observed below the current threshold. We have explained the observation using the schematic shown in **Fig. R11**. As discussed in the main manuscript, the fermions in different Fermi-pockets gain a transverse velocity due to $\Omega_{\mathcal{E}}$. Eq. 1 given in the main manuscript, shows that the magnitude of the current going into different arms of the device from different Fermi-pockets is proportional to $\Delta\Omega_{\mathcal{E}}$. The appearance of the non-linear response after the current threshold suggests that $\Delta\Omega_{\mathcal{E}}$ is non-zero only above it. However, $\Omega_{\mathcal{E}}$ do exist on individual topological Fermi-pockets even when $\Delta\Omega_{\mathcal{E}}$ is zero. The absence of $\Delta\Omega_{\mathcal{E}}$ causes the fermions to scatter equally in both arms of the device from both of the Fermi-pockets. Therefore, the chiral current in both arms would not be preferentially carried by one of the Fermi-pockets. However, the chiral currents in both arms do exist! However, the nonlinear responses of the current of opposite chirality cancel out below the threshold current. The condition when there is no occupational imbalance between two Fermi-pockets in both arms below the threshold current is schematically depicted in **Fig. R11(a)**. $\Delta\Omega_{\mathcal{E}}$ becomes non-zero above the threshold current. Therefore, the fermions from the Fermi-pockets at R and Γ are preferentially scattered into the left and right arms, respectively. It creates an occupational imbalance in these arms, which leads to the observation of the chiral current response from the individual Fermi-pockets. This scenario is depicted in **Fig. R11(b)**. The presence of a chiral current in both scenarios makes it possible to observe the quantum interference of the chiral current even before the threshold current. We have a section in the manuscript on this discussion.

(8) Is there any detailed explanation about how the chiral currents acquire a phase due to the current-induced orbital magnetizations? I strongly recommend that the authors add a schematic to illustrate the mechanism of the current- and field-controlled quantum interference.

$$\theta(r) \propto \int (A + A_{\mathcal{E}} + \dots) \cdot dr$$

Fig. R12 | Schematic showing the phase acquired by the chiral fermion due to the presence of vector potential contribution from the external magnetic field and quantum geometry induced orbital magnetization.

The phase of a charged particle's wavefunction is modified by the electromagnetic potentials, which in our study are primarily orbital magnetization and external magnetic field. As we know in a typical Aharonov-Bohm effect, the electron's phase is modified by the vector potential \mathbf{A} of the external magnetic field \mathbf{B} , which is given as $\nabla \times \mathbf{A}$. Our study on the nonlinear responses and their relative modulation with the external magnetic field showed that the chiral current in both arms carries orbit magnetization of opposite polarities. The orbital magnetization causes the fermionic wavefunction to gain a phase even in the absence of an external magnetic field. The orbital magnetization comes due to the electric-field induced Berry curvature $\boldsymbol{\Omega}_{\mathcal{E}}$, given as $\boldsymbol{\Omega}_{\mathcal{E}} = \nabla \times \mathbf{A}_{\mathcal{E}}$. The electric-field induced Berry connection $\mathbf{A}_{\mathcal{E}}$, given as $g\mathcal{E}$, is mathematically analogous to the vector potential \mathbf{A} of the external magnetic field. We derived the phase acquired by the chiral current due to $\mathbf{A}_{\mathcal{E}}$ in the absence of an external magnetic field in Eq. S8-S10. We have used the assumption of a free-particle wavefunction, which is invalid for multifold fermions experiencing current-induced magnetization. However, Eq. S10 gives a qualitative understanding of the quantum interference of the chiral current in the absence of a magnetic field. As the referee recommended, we have added a schematic showing the phase acquired by the chiral fermion due to orbital magnetization and magnetic fields.

(9) This question is about the reliability of the experimental results. How many PdGa devices have they fabricated and measured? Do they exhibit similar performances in terms of threshold currents and the magnitude of NLH signals? Can they fabricate several quantum interference devices with different arm lengths? If so, they can quantify the coherence length, which is a very important parameter for practical applications.

We kindly ask the reviewer to refer to our response to question 5 of Referee 1, where we discussed various measured devices and their reproducibility. In summary, we measured 23 devices in different geometries (three-armed and two-armed) and crystallographic directions. The devices measured along the same crystallographic orientation and device geometry showed similar performance, namely, the appearance of distinct NLH responses after the threshold current in different arms, and their distinct symmetries with the magnetic field orientation. In device with different device geometry however the same crystal orientation, the threshold current varied from 53 – 58 μA . We have fabricated devices with MZI geometry with different arm lengths ranging from 15 – 25 μm . We observed clear quantum interference signals in these devices. However, the amplitude of the oscillations didn't correlate with arm's length. The reason is that the nonlinear responses due to the quantum metric don't scale with the scattering time, which dictates the Drude resistivity.⁷ The quantification of the coherence length requires knowledge of the decoherence mechanism. As we discussed earlier, the decoherence mechanism of the chiral current is not entirely known. The temperature-dependent study of the oscillation amplitude

correlates the magnitude of the quantum metric. The oscillations persisted even at 3.4 K, if the quantum metric is non-zero and the lock-in could demodulate the $V_{3\omega}$ response from the input signal.

Fig. R13 | (a) False coloured image of the device with two Mach-Zehnder Interferometers (MZI) in series. The crystal axis of the first MZI is rotated 45° with respect to the second MZI. **b(i)** SEM of the MZI made in the same crystal orientation mentioned in the manuscript, however with large arm length. **b(ii)** Oscillations of $V_{3\omega}$ response with applied current of $30 \mu\text{A}$ in the device shown in **b(i)** measured. **c(i)** SEM image of the modified version of the MZI shown in image **b(i)** with a different width-to-thickness ratio. **c(ii)** The oscillation in the third-order response with applied current of $30 \mu\text{A}$ in the device shown in **c(i)**.

We have also studied the role of crystallographic directions of applied current, i.e., quantum geometry, in observing chiral current and their interference responses. As shown in **Fig. R13a**, we fabricated two MZIs in series having different crystallographic directions in the same device. The MZI on the right has the crystal orientation discussed in the manuscript, while the left MZI has a geometry 45° rotated. We could observe the quantum interference of NLH current in the across the right MZI, while the interference responses were not observed in the left MZI. It clearly shows the importance of quantum geometry to achieve quantum interference. Next, we studied the role of the shape anisotropy of the MZI arms in interference. In the MZI shown in **Fig. R13b(i)**, the left and right arms of the device have the same width-to-thickness ratio (close to 1). **Fig. R13b(ii)** shows clear oscillations of $V_{3\omega}$ response with applied current, with an amplitude of around 250 nV. In the same device, we changed the width-to-thickness ratio between different arms as shown in **R13c(i)** using the FIB. The amplitude of the oscillations was significantly reduced, close to 5 nV. Our results demonstrate that tuning the chiral

fermionic valve such that a similar magnitude of chiral current in both arms is important to observe a good quantum interference signal.

Reply to Referee #3:

This manuscript reports a study on chiral fermionic transport in a three-arm device fabricated from the chiral topological semimetal PdGa. The authors fabricate high-quality device using advanced microfabrication techniques (focused-ion-beam milling). Through systematic investigations of second/third-order nonlinear Hall responses and magnetic field dependencies, the authors demonstrate the spatial separation of chiral fermions with opposite chirality into distinct arms of the device. Additionally, they show that the quantum interference of the chiral currents can be observed by using a Mach-Zehnder interferometer, confirming their long-range phase coherence. This work provides novel insights into the design of topological electronic devices based on quantum geometry and holds significant scientific and practical potential. I recommend the manuscript for publication, subject to revisions listed below:

We thank the Referee for the kind remarks, for acknowledging our work's novelty, and for recommending our work for publication. We have now consistently derived the non-linear Hall responses up to third-order in a semiclassical Boltzmann transport formalism, incorporating changes to the band energy and Berry curvature due to the applied electric field. We have also performed a symmetry analysis to determine the allowed responses based on the crystalline symmetries of PdGa. Details are summarised in the Supplementary file and discussed individually with each question.

1. The semiclassical Boltzmann kinetics (Eqs. S1–S6) provides qualitative insights into the nonlinear responses but lacks validation against effective model or first-principles calculations. Incorporating numerical calculations of nonlinear conductivity based on Berry curvature and quantum metric would strengthen the theoretical framework.

The referee raised an important aspect regarding using numerical methods to quantify the nonlinear conductivity based on quantum geometry. The theoretical framework based on the Boltzmann equation and first-principal calculations assumes electrons as non-interacting adiabatic Bloch wavepackets moving in a static band structure under equilibrium. (In response to Referee 1's question 3, we have clearly stated these assumptions.) It is a key limitation to model the transport at nonequilibrium due to topological bands with multifold degeneracies. As discussed in the manuscript, we observed nonlinear responses due to the phase coherent chiral currents only after a certain current threshold. In response to Referee 2's question 1, we have discussed that the transition with current is linked with the intrinsic property of the band structure at nonequilibrium. Therefore, numerical calculations performed on a static band structure at equilibrium conditions cannot model the nonlinear responses observed in our study after the transition at the current threshold. Furthermore, semiclassical Boltzmann kinetics typically assumes a single distribution function per band, effectively treating all carriers as originating from a single Fermi surface. However, we have shown two distinct nonlinear responses coming from two different topological Fermi-pockets. These limitations present an interesting avenue to develop new theoretical model which accounts for electric-field-induced transition in the bandstructure associated metric under nonequilibrium. We believe that a systematic study of accurately modelling these transitions is needed, which is beyond the primary focus of our study.

2. There are several distinct mechanisms of nonlinear transport, each subject to different symmetric constraints and scaling behaviors. However, the authors arbitrarily attribute the observed (2nd and 3rd

order) nonlinear signals to the quantum geometric quantities (g and Ω) without establishing a rigorous connection to crystal symmetry or band structure. The authors should provide further explanation.

We have experimentally and theoretically examined various trivial-state-based mechanisms that can produce a nonlinear Hall response. We kindly request the Referee to refer to our response given to Referee's 1 question 1, where we have discussed possible involvement of spin-galvanic effect due to the spin-splitting of trivial bands. We have then provided experimental evidence to rule it out. We have observed distinct symmetry of anomalous Hall response upon varying the crystallographic direction of applied current. In Fig. R1 and R3, we have shown that the distinct magnitude and symmetry of the chiral current in different arms observed can only be explained by incorporating the different quantum geometry of topological bands. In Fig. R3, we have also established the connection between the symmetry of observed anomalous Hall response with the chiral current response discussed with Fig. 4 of the manuscript. Our ability to control the relative magnitude of chiral current in different arms with crystallographic direction of applied current provides a direct connection between the observed nonlinear signal and the crystal symmetry. Theoretically, we have also considered the contribution of nonlinear responses coming from second and third-order Boltzmann equations. We have systematically derived those equations and provided symmetry arguments to rule them out as the sole contributor to our measured nonlinear responses. Please refer sections 3 and 4 in the supplementary file. We like to point out the even though the symmetry of a crystal is defined by its space group, the symmetry of a Fermi pocket is determined by the local symmetry of the band structure at a given k -point, which can differ from the overall crystal symmetry.¹ We have exploited the distinct symmetries of the topological Fermi pockets at R and Γ to establish the origin of the nonlinear responses due to their quantum geometry.

Fig. R14 | Magnetic field dependence of nonlinear responses. a-b Variation of third-order and second-order responses with applied current at different out-of-plane magnetic fields.

Now we will discuss the scaling behaviour of nonlinear responses with respect to first-order response. **Fig. R14** shows the nonlinear responses measured at different out-of-plane magnetic fields. Meanwhile, **Fig. R7c** shows the first-order response measured simultaneously in the same device. We can observe that the nonlinear responses don't vary with magnetic field, even when the first-order response clearly shows field dependence, especially after the threshold current. The magnetic field dependence of first-order conductivity is expected due to the scattering caused by cyclotron motion. Since the nonlinear responses don't vary with magnetic field, it suggests that the nonlinear conductivity is independent of first-order conductivity. Such scaling behaviour is expected for the band-normalized quantum metric-induced nonlinear responses.⁷

3. The authors devoted considerable words explaining the Fermi surfaces and Chern numbers at different high-symmetry points. However, the lack of both intuitive band structure presentation and explicit Chern number calculations weakens their analysis. Further addition of the first-principles calculations would enhance the clarity of their discussion.

The first-principle calculations to calculate the band structure, Fermi surfaces and Chern number of topological bands of PdGa has been extensively studied in the literature.^{4,14-17} We have added references to these studies in our manuscript.

4. In lines 165-167, the authors declare that PdGa breaks P and T simultaneously, thus attribute the nonlinear Hall signal to the quantum metric g . Moreover, in lines 171-173 the authors claim that a nonzero g is the evidence of T breaking. This claim relies on the very assumption it is meant to prove. Considering PdGa is a non-magnetic material, the authors need to provide more explanations of this point.

We have modified the main text to make our reasoning clearer, which is as follows. Eq. 1 in the manuscript shows that the nonlinear current due OAM dipole $\Delta\Omega_{\mathcal{E}}$ can only be generated in a system with non-zero g . We have experimentally shown in Fig. 2 and 3 of the manuscript that nonlinear responses start to appear after the threshold current. Hence, we have claimed that the g is non-zero after the threshold current. Additionally, we observed the sign reversal second-order responses for chiral current moving into different arms of the device, as shown in Fig. 3. The studies on nonlinear Hall responses due to quantum metric have linked such a sign reversal to the presence of non-zero g .⁷ The non-zero g appears in a system that simultaneously breaks P and T symmetry while maintaining PT symmetry. Therefore, we have argued that the T is locally broken in our systems due to the presence of OAM dipole in the individual arms, which also satisfies the PT symmetry criteria. We relate the non-zero quantum metric g , commonly observed in antiferromagnetic systems, to our system by identifying the OAM dipole as the orbital analogue of the spins in an antiferromagnet.

5. The expressions of nonlinear transport [Eqs. (1), (S6a), and (S6b)] only consider the electric field modified Berry curvature, but the electric field modified band energy is ignored, which affect the Fermi-Dirac distribution f_0 . The absence of this modification gives totally conflict results, especially in symmetric constrains.

The authors need carefully examine the relevant references [PRL 112,16601 (2014), PRL 127,277201 (2021), PRL 127,277202 (2021), NSR nwae334 (2024)].

We have carefully read the papers highlighted by the referee. We thank the referee for suggesting that we incorporate the electric-field-modified band energy. We have incorporated second-order and third-order corrections to the electric field modified distribution function. Please refer to the section 3 in the supplementary file. Most terms due to these corrections either cancel out due to symmetry reasons or they don't compete to negatively influence our experimental results.

6. The authors emphasize the significance of the quantum metric in nonlinear transport throughout the manuscript. However, they have completely confused this concept—in the original paper [PRL 112,16601 (2014)], " g " represents the Berry curvature polarizability (BCP) or the band-normalized quantum metric, not the quantum metric. The authors are strongly advised to carefully reorganize the

theoretical analysis.

We thank the referee for pointing out this important distinction. We have incorporated the suggestion into the manuscript. We have now denoted \mathbf{g} as the band normalized quantum metric in the manuscript, which captures the local geometric properties of Bloch states in momentum space. We have also made a connection between \mathbf{g} and Berry connection polarization tensor \mathbb{P} used in the paper [PRL 112,16601 (2014)].

7. In line 50, the authors refer to the quantum geometric tensor T and provides its expression. Most strikingly, the authors exclusively use tensor arrow notation exclusively for the quantum metric tensor, while completely omitting such notation for the quantum geometric tensor and the Berry curvature tensor. This misleading notation creates unnecessary confusion and raising serious concerns about this manuscript.

We thank the referee for the remarks. We have made the necessary changes in the manuscript to have a uniform notation of the tensors.

8. The manuscript contains numerous notational inconsistencies. For instance, the symbol $\Delta\Omega$ is ambiguously used to denote both field-modified Berry curvature and orbital angular momentum dipole. We strongly recommend the authors to rigorously clarify their notational scheme.

We thank the Referee for identifying this important inconsistency. We have clarified the notation between the two quantities in the manuscript. We have denoted $\Delta\Omega_{\mathcal{E}}$ as the OAM dipole arising due to the field-induced quantum geometry of the topological bands.

9. The manuscript contains numerous nonstandard and erroneous vector notations. According to conventional typographic practices, bold font should denote vectors/tensors while regular font with subscript should indicate components. However, the authors violate this convention (e.g., $\mathbf{v}_a = -e\mathbf{E} \times \Omega$), with similar confusing notation appearing in at least 4 different places. We strongly recommend the authors conduct a thorough revision to rectify all issues throughout the text.

We thank the Referee for the suggestions to improve the vector notations following conventional practices. We have made these changes to ensure consistent notation throughout the manuscript.

10. A series of typos need be fixed. Line 60: A period at the end of the sentence is missing. Line 439: f is not in the formula environment. Line 441: Redundant parentheses.

We thank the Referee for pointing out these errors. We have corrected these typos.

References:

1. Cano, J. & Bradlyn, B. Band Representations and Topological Quantum Chemistry. *Annu. Rev. Condens. Matter Phys.* **12**, 225–246 (2021).
2. Schade, N. B., Schuster, D. I. & Nagel, S. R. A nonlinear, geometric Hall effect without magnetic field. *Proc. Natl. Acad. Sci.* **116**, 24475–24479 (2019).

3. Furuta, S. *et al.* Reconsidering nonlinear emergent inductance: Time-varying Joule heating and its impact on AC electrical response. *Phys. Rev. B* **110**, 174402 (2024).
4. Polatkan, S. & Uykur, E. Optical Response of Chiral Multifold Semimetal PdGa. *Crystals* **11**, 80 (2021).
5. Ganichev, S. D. *et al.* Spin-galvanic effect. *Nature* **417**, 153–156 (2002).
6. Ma, H., Cullen, J. H., Monir, S., Rahman, R. & Culcer, D. Spin-Hall effect in topological materials: evaluating the proper spin current in systems with arbitrary degeneracies. *Npj Spintron.* **2**, 55 (2024).
7. Gao, A. *et al.* Quantum metric nonlinear Hall effect in a topological antiferromagnetic heterostructure. *Science* **381**, 181–186 (2023).
8. Kawamura, M. *et al.* Laughlin charge pumping in a quantum anomalous Hall insulator. *Nat. Phys.* **19**, 333–337 (2023).
9. Kane, C. L. Telegraph Noise and Fractional Statistics in the Quantum Hall Effect. *Phys. Rev. Lett.* **90**, 226802 (2003).
10. Taktak, I. *et al.* Two-particle time-domain interferometry in the fractional quantum Hall effect regime. *Nat. Commun.* **13**, 5863 (2022).
11. Ahmad, A., K., G. V. & Sharma, G. Chiral anomaly induced nonlinear Hall effect in three-dimensional chiral fermions. *Phys. Rev. B* **111**, 035138 (2025).
12. Gilbert, M. J. Topological electronics. *Commun. Phys.* **4**, 70 (2021).
13. Gilbert, M. J. Chern networks: reconciling fundamental physics and device engineering. *Nat. Commun.* **16**, 3904 (2025).
14. Schröter, N. B. M. *et al.* Observation and control of maximal Chern numbers in a chiral topological semimetal. *Science* **369**, 179–183 (2020).
15. Yen, Y. *et al.* Controllable orbital angular momentum monopoles in chiral topological semimetals. Preprint at <http://arxiv.org/abs/2311.13217> (2023).
16. Tang, P., Zhou, Q. & Zhang, S.-C. Multiple Types of Topological Fermions in Transition Metal Silicides. *Phys. Rev. Lett.* **119**, 206402 (2017).

17. Bradlyn, B. *et al.* Beyond Dirac and Weyl fermions: Unconventional quasiparticles in conventional crystals. *Science* **353**, aaf5037 (2016).

Peer Review File II

Manuscript Title: Chiral Fermionic Valve driven by Quantum Geometry

Referees' comments:

Referee #1 (Remarks to the Author):

The revised manuscript includes new experimental data across multiple devices, detailed temperature dependence, additional anomalous Hall measurements, and a discussion of theoretical limitations. The concept of the chiral fermionic valve is now supported by both passive control achieved through crystallographic orientation, and preliminary active control, achieved through integration with magnetic tunnel junctions, MTJs.

The novelty and significance of this work are considerable. The authors introduce a fundamentally new device concept that enables chirality-selective transport control in a topological semimetal, which could have major implications for future quantum electronic devices. The experimental approach is rigorous: multiple harmonic responses (1ω , 2ω , 3ω) are measured systematically, with a clear current threshold behavior and a careful symmetry analysis of the nonlinear Hall effect. The revision expands the dataset significantly, with results from 23 devices and multiple geometries, strengthening the case for reproducibility. Additional anomalous Hall measurements convincingly disfavor trivial-state mechanisms and bolster the claim that the observed effects arise from topological fermions. Importantly, the authors now acknowledge the limitations of their semiclassical modelling, explicitly discuss possible trivial contributions, and mark their discussion of Fermi-arc mediated coherence as a hypothesis. Finally, the inclusion of MTJ-based devices demonstrates a first step toward tunability, providing proof-of-concept for an actively controllable valve.

However, some limitations remain. The demonstration of active control is promising but still preliminary, with no systematic characterization of switching reproducibility, fidelity, or scalability. While the evidence presented makes trivial contributions unlikely, they cannot be strictly ruled out, and the manuscript would benefit from an explicit acknowledgment of this residual ambiguity. The reproducibility of results is now supported by a larger dataset but remains mostly documented in the Supplementary file, while the main text still focuses on single-device examples. Although the temperature dependence of the nonlinear responses and quantum interference is now provided, a quantitative extraction of coherence length or phase-breaking rate is still lacking. Finally, while the semiclassical framework is acknowledged to be limited, the paper would be stronger if some form of effective modelling or numerical estimate were offered to connect the observed signals more quantitatively to the underlying quantum metric and Berry curvature.

The manuscript would benefit from a few final refinements before publication. First, the authors should clearly state in the main text that while trivial-state contributions have been systematically examined and are strongly disfavored, they cannot be mathematically excluded. This would appropriately temper the strength of the claim. Second, the authors should consider including a summary table or figure in the main text quantifying reproducibility across all devices, showing, for example, the threshold current, nonlinear Hall amplitude, and interference visibility, rather than relegating these data to the Supplementary material. Third, the language describing the MTJ-based active control should be softened to emphasize that these experiments represent a proof-of-concept demonstration rather than a fully operational valve technology, and the authors could briefly discuss open questions such as switching speed and reproducibility. Finally, the discussion could briefly highlight the absence of a quantitative coherence length extraction and the need for more advanced theoretical modelling, positioning these as important directions for future work.

The revised manuscript is a significant improvement over the original manuscript. With the clarifications suggested above, it will be a strong candidate for publication in Nature. The combination of chirality separation, orbital magnetization control, and quantum interference in a single platform represents a compelling advance for topological quantum electronics.

We thank the referee for thoroughly reviewing our work and giving us very constructive and encouraging feedback. We have made the following refinements to our manuscript, as suggested by the referee –

- 1) We have now stated in the main text that trivial-state contributions may coexist with the quantum geometry-induced responses; however, they don't account for the observed NLH responses in our study. We have also moved the discussion on the possible trivial state mechanism and temperature dependence to the Method section from the Supplementary file.
- 2) We have added the summary table to the Method section of the manuscript and included an additional table showing the threshold current and NLH signal amplitudes of devices with different dimensions. We have also discussed the interference visibility of the MZI interferometer with different ϕ values in the text.
- 3) We have added a section on active control of the valve clearly stating it as a proof-of-concept demonstration. We have also discussed open questions which needs to be addressed to pursue this direction further.

- 4) We have now discussed in the main text a need for more advanced theoretical study to understand the decoherence mechanism and quantify the coherence length.
- 5) The numerical estimate of the NLH response requires the accurate theoretical modelling of the observed transition after the threshold current. We had discussed in our previous report that the transition is linked with the metric of the bandstructure under non-equilibrium due to applied electric field. We posit that the appearance of g after a certain current threshold is linked with the appearance of current-induced mesoscopic magnetization in the system due to orbital angular moments (OAM). We believe that the long-range magnetization originates from inter-atomic overlap of OAM, rather than the local OAM of single Bloch wavepacket.^{1,2} We can formulate the picture in terms of orbital magnetic susceptibility χ_g originating due to quantum metric.³ χ_g is defined as a degree of (mesoscopic) magnetization in response to an effective magnetic field. The question arises that what is this effective magnetic field and why does it only arise after a certain current threshold? We present one of the possible answers which can explain these phenomena. The presence of current threshold suggests that a certain amplitude of electric-field induced OAM is needed, for it to partake in an overlap with another wavefunction of the neighbouring lattice sites. A finite overlap (or amplitude distance in reciprocal space) between the wavefunctions of neighbouring lattice sites acts like a magnetic field in the form of effective OAM-OAM exchange interaction. Such overlaps between wavefunctions of different lattice sites in series can thus trigger a mesoscopic magnetization. Our analysis presents an intuitive picture behind the possible cause of the observed transition, which needs to be further explored.

Referee #2 (Remarks to the Author):

The revised manuscript has been greatly improved, and the authors' replies have addressed most of my earlier concerns in a constructive manner. The work is of high interest, and I find the central idea of a chiral fermionic valve both original and potentially impactful. Nevertheless, I still have several questions regarding the work's broader impact and technical validation, which need further clarification:

We sincerely thank the referee for the positive remarks about our study.

1. The supplementary figures (e.g., Fig. S4) show the current dependence of first-, second-, and third-order responses, but the expected quadratic ($V_{2\omega} \propto I^2$) and cubic ($V_{3\omega} \propto I^3$) scaling are not explicitly demonstrated. Presenting log-log plots with fitted slopes would provide a clearer validation of the power-law relations and help exclude alternative nonlinear mechanisms.

Fig. R15 | a-b Variation of second-order and third-order responses with applied current along with the fitted polynomial curves.

Fig. R15 a-b show the polynomial fit for $V_{2\omega}$ and $V_{3\omega}$ responses with applied current. We can observe that the fit is significantly better for the data above the threshold current, while some additional features are evident below it. The log-log plots were not feasible for negative y-axis values. The fit below the threshold current is poor because the transition observed in our study cannot be captured by polynomial fits alone. We discussed a possible explanation for this transition in our response to Referee 1 (point 5). We emphasize that further theoretical analysis is needed to quantify our NLH data, which presents an important direction for future work.

2. This paper demonstrates a novel device concept with potential applications in "topological quantum electronics". It would be highly valuable for the authors to elaborate on and discuss more concrete application scenarios. For example, could this valve structure serve as a fundamental building block for a specific type of memory or logic gate, a highly sensitive magnetic/orbital field sensor, or a component in a quantum information processing architecture?

We thank the Referee for asking these important questions regarding the potential application of the chiral fermionic valve towards topological quantum electronics. We believe that one of the immediate directions to explore would be towards cryogenic memory applications. In our discussion with the active control of the valve, chiral currents in both arms are sensitive to the magnetization direction of the MTJ. Additionally, in our preliminary results, we have also seen switching of the magnetization of the MTJ with chiral current. These functionalities are crucial to realize cryogenic memories, which would also be useful for quantum computation. The other use cases which the Referee suggested are also very intriguing, which we will definitely explore in our future studies.

3. The authors mention that the underlying mechanism is almost independent of relaxation time, which enables the observation of quantum interference over a length scale exceeding 15 μm . From a fundamental perspective, a critical question emerges: What is the expected scalability of this phenomenon? Is there a theoretical upper limit to the device size in which this coherence can be sustained? Discussing whether this chiral transport and interference could, in principle, exist on a macroscopic scale would provide profound insight into the robustness and ultimate boundaries of the observed quantum effects.

The referee raised an important question about the fundamental limit of the length scale of our observed effect. We posit that the scalability of our observed phenomenon depends on the crystal quality. Our samples were fabricated from a single crystal; thus, the information regarding the quantum geometry can be extracted at a mesoscopic scale. However, we speculate that this property would be lost in a thin film of the same material with multi-domains. Secondly, we have postulated the role of the Fermi arcs in chirality coherence in the manuscript. It proposes that the transverse electric field of opposite sign is present in each of the arms, which prevents the chiral charge leakage between the two topological Fermi pockets. We expect that the coherence length should decrease upon increasing the thickness or width of the arm, as the transverse electric field would diminish in magnitude.

Referee #3 (Remarks to the Author):

I am generally satisfied with the authors' revisions and recommend the manuscript for publication after several minor revisions. Specifically, there remain a few points that need to be commented on to enhance the legibility of the manuscript.

We are thankful to the referee for recommending our manuscript for publication.

1. Orbital angular momentum (OAM). The current discussion of OAM is vague, and its mathematical relation to quantum geometric quantities has not been clarified. Given the central role of OAM in the manuscript, I strongly suggest the authors provide a better argument.

The modern theory of polarization⁴ can describe OAM of Bloch bands in terms of quantum geometric quantities. We first consider an effective two-dimensional Bloch Hamiltonian in the spinor basis as follows

$$\hat{H}(k) = \begin{pmatrix} \hat{h}_\uparrow(\mathbf{k}) & 0 \\ 0 & \hat{h}_\downarrow(\mathbf{k}) \end{pmatrix}$$

The orbital moment along z for a band α with spin σ is defined as^{5,6}

$$\ell_{\alpha\sigma}^z(\mathbf{k}) = \frac{m}{\hbar} \text{Im} \langle \partial_{k_z} u_{\mathbf{k}\alpha\sigma} | \hat{h}_\sigma(\mathbf{k}) - \varepsilon_{\mathbf{k}\alpha} | \partial_{k_y} u_{\mathbf{k}\alpha\sigma} \rangle$$

The resultant orbital magnetization is given as $m_z(\mathbf{k}) = (e/m)\ell_{\alpha\sigma}^z(\mathbf{k})$. The equation given above is composed of a term proportional to the Berry curvature, which is given as follows^{5,6}

$$\ell_{\nu\sigma}^z(\mathbf{k}) = -\frac{m}{\hbar} (\varepsilon_{\mathbf{k}c} - \varepsilon_{\mathbf{k}v}) \Omega_{\nu\sigma}(\mathbf{k})$$

where $\varepsilon_{\mathbf{k}c}$ and $\varepsilon_{\mathbf{k}v}$ are the eigenvalue of $\hat{h}_\sigma(\mathbf{k})$ for \mathbf{k} in the conduction and valence band, respectively. In the presence of non-zero quantum metric, we would also incorporate the electric field-induced Berry curvature. Therefore, the above equation would be modified as,

$$\ell_{\nu\sigma}^z(\mathbf{k}) = -\frac{m}{\hbar} (\varepsilon_{\mathbf{k}c} - \varepsilon_{\mathbf{k}v}) (\Omega_{\nu\sigma}(\mathbf{k}) + \nabla_{\mathbf{k}} \times (\mathbf{G}\mathcal{E}))$$

Thus, the orbital magnetization originating due to electric-field OAM constitutes a term proportional to Berry curvature and quantum metric. We have now incorporated the references discussing these equations in the manuscript. It is to be noted that the description given above assumes the localized property of the Bloch

wavepacket. Further, analysis needs to be performed to calculate the magnetization due to mesoscopic phase-coherent chiral current.

2. Time-reversal symmetry and quantum metric. In my previous review, I raised concerns regarding the relation between time-reversal symmetry (TRS) and the quantum metric. The existence of a finite quantum metric does not depend on the breaking of TRS, while time-reversal invariant systems can still host nonzero quantum metric. What requires TRS breaking (in addition to inversion breaking) is the manifestation of intrinsic nonlinear Hall effects induced by the quantum metric. The authors' reasoning that a finite quantum metric itself is evidence of TRS breaking needs to be revised.

We agree with the referee's comment and have revised our reasoning in the manuscript.

3. Mechanism of nonlinear Hall effect. The attribution of the observed nonlinear Hall response to the quantum metric remains insufficiently discussed. The authors are encouraged to provide at least a more comprehensive discussion on possible mechanisms of the nonlinear Hall effect coexisting with the quantum metric, such as those in arXiv:2410.04995.

We thank the referee for sharing this insightful paper regarding the possible coexistence of disorder-based mechanisms, which share the same symmetry criteria as the quantum metric-induced responses. We have carefully read the manuscript and recognised that the additional five disorder-based mechanisms depend on the Drude scattering time. From Table IV of Appendix A of the manuscript arXiv:2410.04995, we can notice that the quantum metric dipole based NLH response is the only mechanism which is independent of scattering time. We had discussed in our previous referee report that the our observed NLH responses are also independent of scattering time, which sets the Drude conductivity. As referee 2 also pointed out, it is one of the key reasons why we could observe phase-coherent transport of NLH current at a mesoscopic scale in a bulk metal. Nevertheless, we agree with the referee that the extrinsic mechanism can coexist with quantum metric-induced NLH current. However, they don't account for the NLH responses observed in our study. We have now recognized the possible coexistence of such mechanisms in the main manuscript.

Reference

1. Furukawa, T., Watanabe, Y., Ogasawara, N., Kobayashi, K. & Itou, T. Current-induced magnetization caused by crystal chirality in nonmagnetic elemental tellurium. *Phys. Rev. Research* **3**, 023111 (2021).
2. Yang, Q. *et al.* Monopole-like orbital-momentum locking and the induced orbital transport in topological chiral semimetals. *Proc. Natl. Acad. Sci. U.S.A.* **120**, e2305541120 (2023).
3. Piéchon, F., Raoux, A., Fuchs, J.-N. & Montambaux, G. Geometric orbital susceptibility: Quantum metric without Berry curvature. *Phys. Rev. B* **94**, 134423 (2016).
4. Xiao, D., Chang, M.-C. & Niu, Q. Berry phase effects on electronic properties. *Rev. Mod. Phys.* **82**, 1959–2007 (2010).
5. Lee, J. M. Universal intrinsic orbital dynamics from Berry curvature in electronic two-band systems. *Phys. Rev. B* **112**, 054441 (2025).
6. Schüler, M. *et al.* Local Berry curvature signatures in dichroic angle-resolved photoelectron spectroscopy from two-dimensional materials. *Sci. Adv.* **6**, eaay2730 (2020).